# SRSF1-dependent nuclear export inhibition of *C9ORF72* repeat transcripts prevents neurodegeneration and associated motor deficits

Guillaume M. Hautbergue[1,*,**], Lydia M. Castelli[1,*], Laura Ferraiuolo[1,*], Alvaro Sanchez-Martinez[2], Johnathan Cooper-Knock[1], Adrian Higginbottom[1], Ya-Hui Lin[1], Claudia S. Bauer[1], Jennifer E. Dodd[1], Monika A. Myszczynska[1], Sarah M. Alam[2], Pierre Garneret[1], Jayanth S. Chandran[1], Evangelia Karyka[1], Matthew J. Stopford[1], Emma F. Smith[1], Janine Kirby[1], Kathrin Meyer[3], Brian K. Kaspar[3], Adrian M. Isaacs[4], Sherif F. El-Khamisy[5], Kurt J. De Vos[1], Ke Ning[1], Mimoun Azzouz[1], Alexander J. Whitworth[2,**] & Pamela J. Shaw[1,**]

Hexanucleotide repeat expansions in the *C9ORF72* gene are the commonest known genetic cause of amyotrophic lateral sclerosis and frontotemporal dementia. Expression of repeat transcripts and dipeptide repeat proteins trigger multiple mechanisms of neurotoxicity. How repeat transcripts get exported from the nucleus is unknown. Here, we show that depletion of the nuclear export adaptor SRSF1 prevents neurodegeneration and locomotor deficits in a *Drosophila* model of C9ORF72-related disease. This intervention suppresses cell death of patient-derived motor neuron and astrocytic-mediated neurotoxicity in co-culture assays. We further demonstrate that either depleting SRSF1 or preventing its interaction with NXF1 specifically inhibits the nuclear export of pathological *C9ORF72* transcripts, the production of dipeptide-repeat proteins and alleviates neurotoxicity in *Drosophila*, patient-derived neurons and neuronal cell models. Taken together, we show that repeat RNA-sequestration of SRSF1 triggers the NXF1-dependent nuclear export of *C9ORF72* transcripts retaining expanded hexanucleotide repeats and reveal a novel promising therapeutic target for neuroprotection.

[1] Sheffield Institute for Translational Neuroscience, Department of Neuroscience, University of Sheffield, 385a Glossop Road, Sheffield S10 2HQ, UK. [2] MRC Mitochondrial Biology Unit, University of Cambridge, Cambridge Biomedical Campus, Hills Road, Cambridge CB2 0XY, UK. [3] Nationwide Children's Research Institute, Department of Pediatrics, The Ohio State University, 700 Children's Drive, Rm. WA3022, Columbus, Ohio 43205, USA. [4] Department of Neurodegenerative Disease, UCL Institute of Neurology, London WC1N 3BG, UK. [5] Krebs Institute, Department of Molecular Biology and Biotechnology, University of Sheffield, Firth Court, Sheffield S10 2TN, UK. * These authors contributed equally to this work. ** These authors jointly supervised this work. Correspondence and requests for materials should be addressed to G.M.H. (email: g.hautbergue@sheffield.ac.uk) or to A.J.W. (email: a.whitworth@mrc-mbu.cam.ac.uk).

Amyotrophic lateral sclerosis (ALS) and frontotemporal dementia (FTD) are fatal adult-onset neurodegenerative diseases, which respectively cause progressive death of motor neurons with escalating failure of the neuromuscular system and characteristic alterations of cognitive function and personality features. Neuroprotective treatment options are currently extremely limited and the anti-glutamatergic agent riluzole prolongs survival in ALS patients by only approximately 3 months. The most commonly identified genetic cause of ALS and FTD involves polymorphic repeat expansions, composed of hundreds to thousands of the GGGGCC hexanucleotide-repeat sequence (hereafter abbreviated G4C2) in the first intron of the C9ORF72 gene, with autosomal dominant inheritance and incomplete penetrance[1–4]. The repeat DNA sequences are bi-directionally transcribed leading to the characteristic formation of G4C2-sense and C4G2-antisense RNA foci both in ALS and FTD cases[5,6]. The expression levels and splicing of transcripts involved in multiple cellular pathways are affected in ALS models and human post-mortem tissues leading to dysregulation of RNA metabolism, mitochondrial dysfunction, oxidative stress, excitotoxicity, apoptosis, altered mechanisms of autophagy, protein clearance, axonal transport and motor neuron-astrocyte cross-talk (for reviews, see references[1,3]). Consistent with this, widespread alterations of alternative splicing ($>$8,000) and alternative polyadenylation site usage ($>$1,000) were recently identified in biosamples of cerebellum from C9ORF72-ALS patients[5]. We have also reported that alteration of splicing consistency correlates with faster disease progression in C9ORF72-ALS cases independently of the DNA repeat expansion length[7].

The pathophysiology mediated by C9ORF72 repeat expansions potentially involves three extensively-studied mechanisms which may all contribute to neuronal injury and disease progression: (i) RNA toxic gain-of-function by sequestration of RNA-binding factors[8–12]; (ii) protein toxic gain-of-function due to repeat-associated non-ATG (RAN) translation that occurs in all sense and antisense reading frames to produce five dipeptide-repeat proteins (DPRs)[6,13–16]; (iii) haploinsufficiency due to decreased expression of the C9ORF72 protein[2,17,18] which has recently been shown to play a key role in the Rab GTPase-dependent regulation of autophagy[19–21]. We refer to references[22–26] for recent reviews on the mechanisms of C9ORF72-mediated neurotoxicity.

The splicing of the first intron of C9ORF72 does not appear to be affected by the presence of the hexanucleotide repeat expansions as the proportion of unspliced transcripts measured by the exon1–intron1 junction remains similar in control and patient-derived neurons or post-mortem brain tissues[27]. A small proportion of C9ORF72 repeat transcripts retaining pathological repeat expansions in intron-1 escape nuclear retention mechanisms and were detected in the cytoplasm of patient-derived lymphoblasts[28] where they can subsequently be translated into DPRs. Interestingly, nucleocytoplasmic transport defects of proteins and RNA were recently highlighted in Drosophila, yeast and human neuronal models of C9ORF72-related ALS[29–32]. In particular, a loss-of-function screen in Drosophila identified ALYREF (Aly/REF export factor) and NXF1 (nuclear export factor 1), two components of the mRNA nuclear export machinery, as modifiers of the neurotoxicity mediated by C9ORF72 repeat expansions[30]. However, the mechanism(s) driving the specific nuclear export of pathological intron-retaining C9ORF72 repeat transcripts remain to be elucidated. We and others have reported direct binding and sequestration of the nuclear export adaptor proteins ALYREF[33] and SRSF1 (serine/arginine-rich splicing factor 1)[34] onto G4C2-repeat transcripts[11,12]. Our previous research showed that nuclear export adaptors, which directly interact with RNA and the

nuclear export receptor NXF1, remodel NXF1 in an open conformation in concert with subunits of the TREX (Transcription-Export) complex to increase its affinity for mature mRNAs and trigger the process of mRNA nuclear export[35–39]. The remodelling of NXF1 offers a control mechanism to retain unprocessed transcripts in the nucleus[37,40]. Knockdowns of ALYREF in Caenorhabditis elegans[41] and Drosophila melanogaster[42] are dispensable to the global nuclear export of mRNA and development. Notably, only a partial block of nuclear export is induced upon ALYREF depletion in human cells[43], suggesting the existence of other proteins with redundant export function. Consistent with this, multiple conserved human nuclear export adaptors were found to interact with the RNA-binding domain of NXF1 (refs 34,37,44,45) and can simultaneously interact with the same mRNA molecules[44]. A recent study further showed that whilst each of the seven SR-rich splicing factors SRSF1-7 bind thousands of transcripts, the nuclear export of only a small proportion of transcripts ($<$0.5–2% mRNAs) is affected upon individual knockdown of the SRSF1-7 proteins, clearly highlighting redundancy and/or cooperation in the NXF1-dependent nuclear export adaptor function[46]. We refer to references[47,48] for recent reviews on TREX and the nuclear export of mRNA.

Here, we hypothesized that: (i) Excessive binding of nuclear export adaptor(s) onto C9ORF72 repeat transcripts might force interactions with NXF1 and override the nuclear retention mechanisms; (ii) depletion of sequestered export factors that may inappropriately license the nuclear export of intron-retaining repeat transcripts might in turn confer neuroprotection. We used an established Drosophila model of C9ORF72-related disease which exhibits both neurodegeneration and locomotor deficits[16] to identify potential nuclear export adaptor(s) involved in the nuclear export of C9ORF72 repeat transcripts. We also used a combination of neuronal N2A cells and ALS patient-derived neurons and astrocytes to validate our in vivo findings and dissect the molecular mechanisms driving the nuclear export of repeat transcripts and their associated neurotoxicity. In this study, we demonstrate that sequestration of SRSF1 onto C9ORF72 repeat transcripts triggers their NXF1-related nuclear export independently of splicing which leads to the subsequent RAN translation of neurotoxic levels of DPRs. Moreover, we show that the partial depletion of SRSF1 does not alter expression level, intron-1 splicing or nuclear export of the wild-type C9ORF72 transcripts while it specifically prevents C9ORF72-mediated neurodegeneration and in vivo associated motor deficits.

## Results

**ALYREF and SRSF1 directly bind G4C2 and C4G2 repeat RNA.**
We performed in vitro UV-crosslinking assays using purified recombinant proteins and synthetic G4C2x5 and C4G2x5 RNA probes to investigate direct protein:RNA interactions. Recombinant hexa-histidine-tagged human ALYREF, SRSF1 amino-acids 11-196 which retains wild-type ability to bind RNA and NXF1 (ref. 36), and MAGOH, a control protein which does not bind RNA[49], were purified by ion metal affinity chromatography in high salt to disrupt potential interactions with bacterial RNA. Purified proteins were incubated with 5′-end $^{32}$P-radiolabelled G4C2x5 (Fig. 1a) or C4G2x5 (Fig. 1b) RNA probes prior to irradiation with UV where indicated ($+$) and resolved by SDS–PAGE. As shown on the Phosphoimages, covalently-bound RNA molecules remained associated with ALYREF and SRSF1 visualized on the Coomassie-stained panels during the denaturing electrophoresis. These data demonstrate direct interactions of ALYREF and SRSF1 with both sense and antisense repeat RNA in agreement with our previous

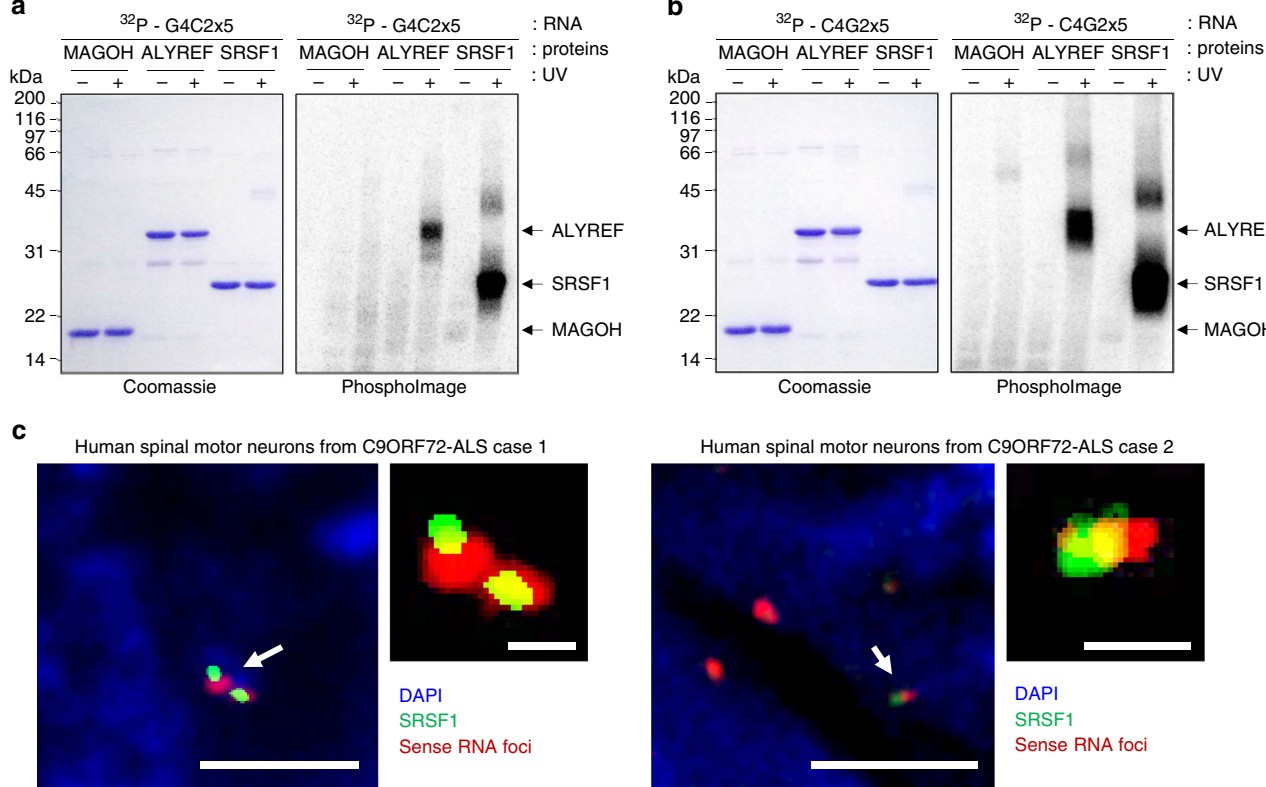

**Figure 1 | Purified ALYREF and SRSF1 proteins directly interact with hexanucleotide repeat sense and antisense RNA.** (**a,b**) Protein:RNA UV crosslinking assays using purified recombinant proteins and 32P-end-radiolabelled G4C2x5 (**a**) and C4G2x5 (**b**) repeat RNA probes. Proteins are visualized on SDS–PAGE stained with Coomassie blue (left panels) and covalently linked RNA:protein complexes by autoradiography on PhosphoImages (Right panels). UV exposure is indicated by $+$. (**c**) Fluorescence confocal microscopy images show co-localization of SRSF1 (labelled in green by immunofluorescence) and sense RNA foci (labelled in red using fluorescence *in situ* hybridization) in motor neurons from post-mortem spinal cord tissues of two human C9ORF72-ALS cases. Nuclei are stained in blue by DAPI. Scale bars in left panels: 3 µm; scale bars in zoomed panels: 0.5 µm.

studies[12,50]. The interactions are specific since no binding of RNA was detected in absence of UV-irradiation or with the negative control protein MAGOH. These direct interactions are also consistent with our previously reported co-localization of ALYREF with RNA foci in motor neurons from C9ORF72-ALS patients[12]. Furthermore, we show that SRSF1 co-localizes with RNA foci in motor neurons from human post mortem spinal cord tissue of C9ORF72-ALS cases (Fig. 1c).

**Depletion of SRSF1 rescues a *Drosophila* model of C9ORF72-ALS.** To gain functional insight into the nuclear export dependence of G4C2-repeat transcripts, we tested whether reducing the expression levels of the conserved nuclear export adaptors ALYREF and SRSF1 might rescue neurodegeneration in C9ORF72-ALS *Drosophila*[16]. Flies expressing 36 uninterrupted repeats ('pure' G4C2x36) were crossed with two independent transgenic RNAi lines each targeting SRSF1 or ALYREF[51] (inverted repeat sequences available in Supplementary Note 1). Targeted expression of G4C2x36 disrupts the compound eye and is minimally altered by co-expression of a control RNAi (Fig. 2a). In contrast, co-expression of two different *SRSF1*-RNAi sequences (also called *SF2/ASF*) completely prevented neurodegeneration, while two knockdown lines of *ALYREF* (also called *Ref1*) only showed a modest rescue of the neurodegenerative phenotype. We validated the successful knockdown of *SRSF1* and *ALYREF* in the corresponding flies showing 70-80% reduction in mRNA expression levels (Fig. 2b). Moreover, the neuronal expression of the G4C2x36 repeat expansion was shown to cause both larval and adult locomotor

deficits in this model[16]. Consistent with a pathogenic role for SRSF1, its partial depletion restored locomotor function in larvae (Fig. 2c) and adult flies (Fig. 2d). This effect is specific of SRSF1 since depletion of ALYREF showed no effect, which is in agreement with the rough eye phenotypes. The neurotoxicity effects observed in the G4C2x36 C9ORF72-ALS model of *Drosophila* were primarily attributed to the expression of DPRs[16]. Accordingly, we now show that the depletion of SRSF1 leads to prominent reduction in the production of both sense and antisense poly-GP DPRs (Fig. 2e).

To test for the hexanucleotide-repeat expansion specificity of the *SRSF1*-RNAi rescued phenotypes, we used the previously established GR36 and PR36 flies[16] which respectively express 36-repeat poly-Gly-Arg and poly-Pro-Arg DPRs using alternative codons. As reported in the original study[16], the GR36 flies have a high rate of mortality and only a few GR36 flies crossed with Ctrl or *SRSF1*-RNAi survived to adulthood. Nonetheless, the partial depletion of SRSF1 did not significantly ameliorate the rough eye phenotypes (Fig. 2f) or the locomotor deficits (Fig. 2g) induced by the G4C2-independent expression of DPRs in both GR36 and PR36-expressing flies. These results indicate that partial depletion of SRSF1 exerts neuroprotection through direct effects on the *C9ORF72* hexanucleotide repeat expansion rather than indirect effect on gene expression alteration or downstream accumulation of DPRs.

**SRSF1 depletion mitigates astrocyte-mediated neurotoxicity.** To apply our *in vivo* findings to human C9ORF72-related ALS, we sought to deplete SRSF1 in patient-derived cell models. Human *SRSF1*-knockdown plasmids co-expressing a GFP

reporter and a pre-miRNA cassette were engineered to produce recombinant *SRSF1*-RNAi lentivirus (LV-*SRSF1*-RNAi) (Supplementary Fig. 1a). HEK293T cells co-transfected with *SRSF1*-RNAi constructs and a FLAG-tagged SRSF1 expression plasmid showed efficient and specific depletion of SRSF1

(Fig. 3a). The survival and morphology of C9ORF72-ALS patient-derived neurons is indistinguishable from control neurons[9,10]. We thus assessed motor neuron survival in co-cultures with patient-derived astrocytes using our recently developed assay that recapitulates the astrocyte-mediated

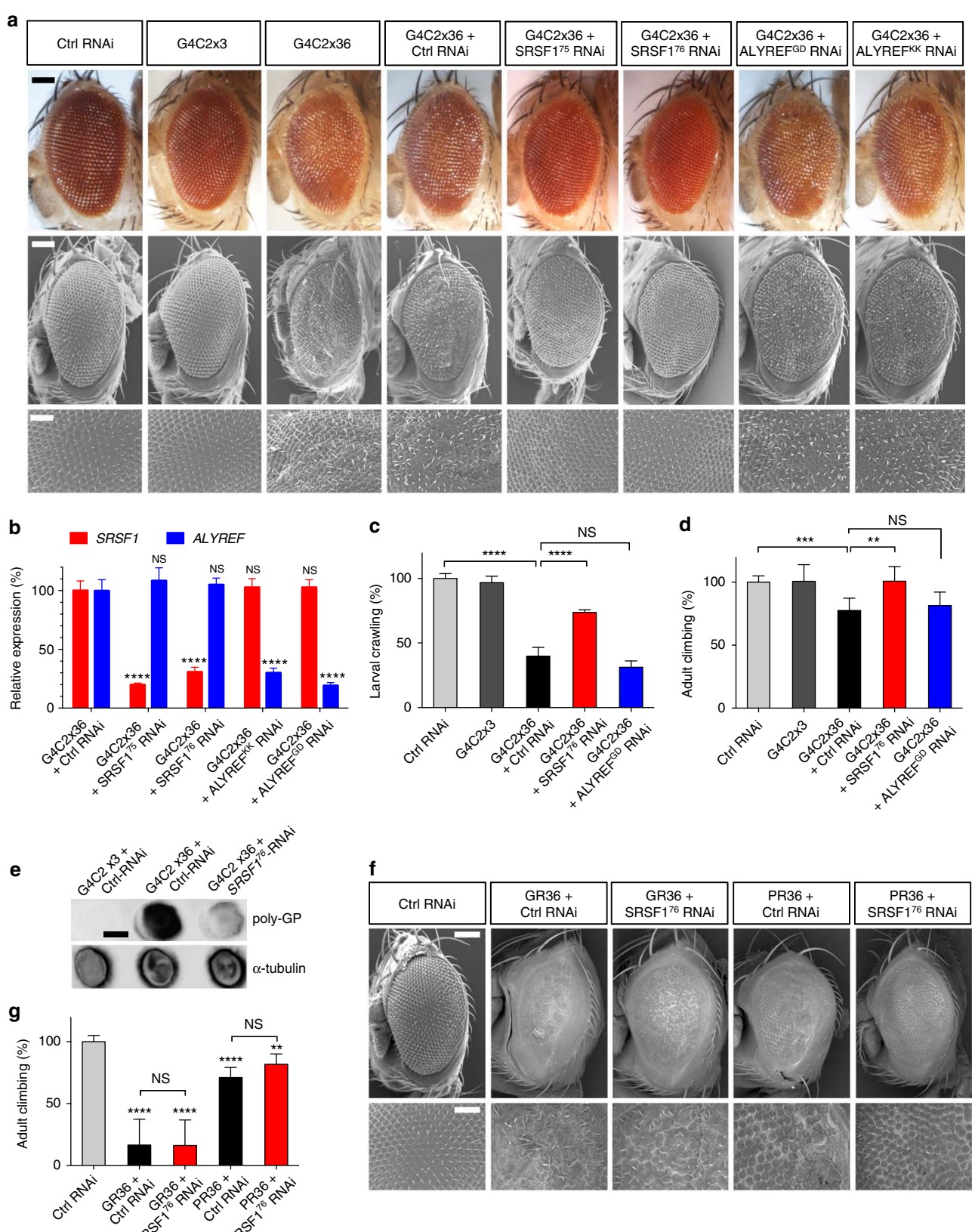

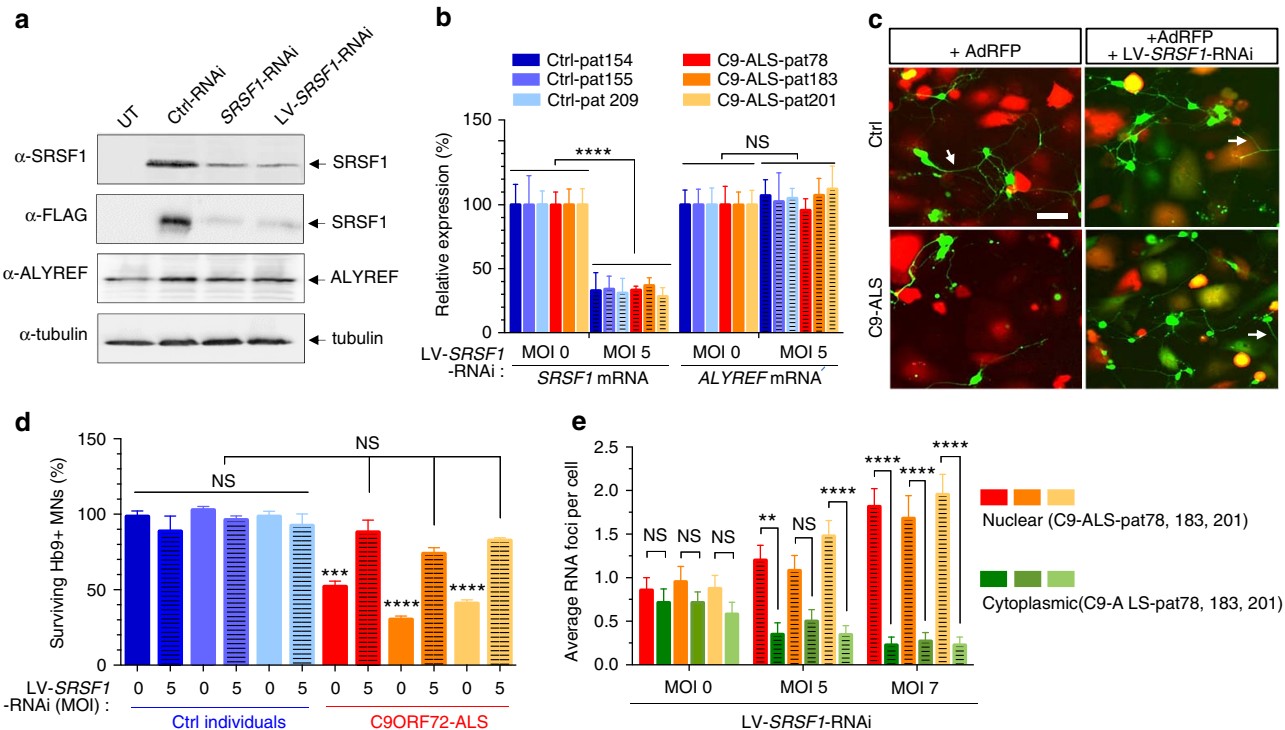

**Figure 3 | Depletion of SRSF1 suppresses patient-derived C9ORF72-mediated astrocytic toxicity and motor neuron death.** (**a**) Untransfected HEK293T (UT), control (Ctrl)-RNAi, SRSF1-RNAi or LV-SRSF1-RNAi and FLAG-tagged SRSF1 transfected cells were analysed 72 h post-transfection by immunoblotting using anti-FLAG, anti-ALYREF and loading control anti-α-Tubulin antibodies. (**b**) SRSF1 and ALYREF transcript levels were quantified by qRT–PCR analysis following normalization to U1 snRNA levels in three biological replicate experiments (mean ± s.e.m.; two-way ANOVA with Tukey's correction for multiple comparisons; N (qRT–PCR reactions) = 6). Three control (Ctrl-pat) and three C9ORF72-ALS (C9-ALS-pat) colour-coded patient lines were used. (**c**) Representative immunofluorescence microscopy images from iAstrocytes and motor neuron co-cultures. iAstrocytes derived from controls (top left image) and C9ORF72-ALS patients (bottom left image) were transduced with Ad-RFP as viral control and cultured with Hb9GFP + motor neurons (green). C9ORF72-ALS iAstrocytes exhibit toxicity against motor neurons as previously reported. Transduction of control Ad-RFP iAstrocytes with LV-SRSF1-RNAi co-expressing GFP led to consistent survival of Hb9GFP motor neurons on control astrocytes (top right image) and increase in survival of motor neurons on C9ORF72-ALS astrocytes (bottom right image). Arrows point to examples of axons of motor neurons. Scale bar, 50 μm. (**d**) Same control (Ctrl-pat) and C9ORF72-ALS (C9-ALS-pat) colour-coded patient lines (as in **b**) are used for quantification of Hb9-GFP + motor neuron counts in four biological replicates of co-cultures of astrocytes and motor neurons at day 3 (mean ± s.e.m.; one-way ANOVA with Tukey's correction for multiple comparisons; N (Hb9-MNs) = Ctrl-pat154: 567/589/582/500, Ctrl-pat154 + SRSF1-RNAi: 620/543/504/349, Ctrl-pat155: 602/610/553/571, Ctrl-pat155 + SRSF1-RNAi: 554/584/532/516, Ctrl-pat209: 519/599/584/535, Ctrl-pat209 + SRSF1-RNAi: 617/486/425/572, C9-ALS-pat78: 352/279/294/258, C9-ALS-pat78 + SRSF1-RNAi: 569/451/398/583, C9-ALS-pat183: 200/188/154/145, C9-ALS-pat183 + SRSF1-RNAi: 480/420/380/399, C9-ALS-pat201: 201/243/261/224, C9-ALS-pat201 + SRSF1-RNAi: 486/463/444/485). (**e**) Quantification of nuclear and cytoplasmic sense RNA foci in SRSF1-RNAi-transduced iAstrocytes (mean ± s.e.m.; two-way ANOVA; N (cells with 1-5 RNA foci) = C9-ALS-pat78 + MOI0: 21, C9-ALS-pat78 + MOI5: 20, C9-ALS-pat78 + MOI7: 22, C9-ALS-pat183 + MOI0: 21, C9-ALS-pat183 + MOI5: 24, C9-ALS-pat183 + MOI7: 22, C9-ALS-pat201 + MOI0: 24, C9-ALS-pat201 + MOI5: 23, C9-ALS-pat201 + MOI7: 22). >95% of cells with RNA foci presented a total of 5 or fewer foci. Statistical significance of data is indicated as follows: NS: non-significant, $P \geq 0.05$; *$P < 0.05$; **$P < 0.01$; ***$P < 0.001$; ****$P < 0.0001$.

**Figure 2 | Depletion of SRSF1 prevents *in vivo* neurodegeneration and restores locomotor function.** (**a**) Representative light and scanning electron micrographs show normal eye phenotypes for Control (GAL4/luciferase-RNAi) and G4C2x3 flies. Expression of G4C2x36 causes a rough eye phenotype which is fully rescued by SRSF1 knockdown. Scale bars: 100 μm for low magnification and 50 μm for zoomed micrographs. (**b**) SRSF1 (red) and ALYREF (blue) transcript levels were quantified by qRT–PCR analysis in independent knockdown lines in G4C2x36 flies. Tub84b transcript levels were used for normalization in three biological replicate experiments (mean ± s.e.m.; two-way ANOVA with Tukey's correction for multiple comparisons; N (qRT–PCR reactions) = 6). (**c,d**) Neuronal expression of G4C2x36 causes larval crawling (**c**) and adult climbing (**d**) deficits that are both restored by SRSF1 depletion (mean ± 95% CI normalized to control; Kruskal–Wallis non-parametric test with Dunn's correction for multiple comparisons; N (larvae) = 10; N (adults) = control (GAL4/luciferase-RNAi): 93, G4C2x3: 26, G4C2x36 + Ctrl-RNAi: 62, G4C2x36 + SRSF1-RNAi: 50, G4C2x36 + ALYREF-RNAi: 36). (**e**) Total protein extracts from G4C2x3 + Ctrl-RNAi, G4C2x36 + Ctrl RNAi and G4C2x36 + SRSF1-RNAi Drosophila larvae were analysed by dot blots using poly-GP and loading control α-tubulin antibodies. Scale bar, 6 mm. (**f**) SRSF1 depletion in Drosophila models expressing DPRs independently of G4C2 repeat expansions and RAN translation16. (**g**) Neuronal expression of poly-Gly-Arg DPRs (GR36) and poly-Pro-Arg DPRs (PR36) causes adult climbing deficits that are not restored by SRSF1 depletion (mean ± 95% CI normalized to Control; Kruskal–Wallis non-parametric test with Dunn's correction for multiple comparisons; N = Control (GAL4/luciferase-RNAi): 239, GR36 + Ctrl-RNAi: 12, GR36 + SRSF1-RNAi: 7, PR36 + Ctrl-RNAi: 125, PR36 + SRSF1-RNAi: 119). Statistical significance of data is indicated as follows: NS: non-significant, $P \geq 0.05$; *$P < 0.05$; **$P < 0.01$; ***$P < 0.001$; ****$P < 0.0001$.

neurotoxicity observed in ALS for both primary mouse and human derived neurons[52].

Transduction of human iNPC (induced neural progenitor cells)-differentiated astrocytes (iAstrocytes) using a viral multiplicity of infection (MOI) of 5 led to efficient transcript knockdown comparable to levels achieved *in vivo* in neuro-protected G4C2x36 + SRSF1-RNAi *Drosophila* (Supplementary Fig. 2a). Mouse GFP-Hb9 + motor neurons were plated onto LV-SRSF1-RNAi transduced astrocytes derived from three controls and three C9ORF72-ALS patient fibroblast lines (Table 1) and automatically counted daily for three days using a high-throughput imaging system (Supplementary Fig. 2b, Methods). Quantification of SRSF1 and ALYREF mRNA levels confirmed the specific and partial knockdown of SRSF1 transcripts in both control and C9ORF72-ALS iAstrocytes (Fig. 3b). Control iAstrocytes transduced with an RFP-adenovirus (red) efficiently supported the growth of GFP-Hb9 + motor

neurons (green) while fewer motor neurons survived when co-cultured with astrocytes derived from C9ORF72-ALS patient fibroblasts[52] (Fig. 3c). Quantification of surviving motor neurons from four replicate experiments showed that while depletion of SRSF1 is not detrimental to control co-cultures, motor neuron death was prevented by depletion of SRSF1 in co-cultures derived from three separate C9ORF72-ALS cases (Fig. 3d).

To investigate potential nuclear export alterations of G4C2 repeat transcripts, we quantified nuclear and cytoplasmic sense RNA foci in the C9ORF72-ALS iAstrocytes in blinded experiments. Representative images and individual counts are respectively reported in Supplementary Fig. 3 and Supplementary Table 1. Upon depletion of SRSF1, the average number of foci per cells remained similar. However, the number of cytoplasmic RNA foci decreased while nuclear RNA foci concomitantly increased upon SRSF1 depletion (Fig. 3e) consistent with a potential nuclear export inhibition of G4C2-repeat transcripts.

**RAN-dependent translation of DPRs in neuronal cell models.** To investigate whether the binding of SRSF1 to G4C2-sense and C4G2-antisense repeat RNA sequences has the ability to trigger the nuclear export of repeat transcripts, we generated synthetic mammalian expression constructs bearing increasing lengths of pure repeat sequences in the absence of ATG or Kozak elements to specifically investigate RAN-dependent translation of dipeptide repeat proteins. Following annealing of synthetic G4C2 or C4G2 repeat oligonucleotides as described in Methods and Supplementary Fig. 4a–d, we engineered plasmids expressing transcripts containing 15 or 38 uninterrupted sense repeats (G4C2x15 or G4C2x38) and 15 or 39 uninterrupted antisense

**Table 1 | List and characteristics of patient-derived cells used in this study.**

| Patient sample | Ethnicity | Gender | Cell type | Age at biopsy collection |
|---|---|---|---|---|
| 78 | Caucasian | Male | C9ORF72 | 66 |
| 183 | Caucasian | Male | C9ORF72 | 49 |
| 201 | Caucasian | Female | C9ORF72 | 66 |
| 154 | Caucasian | Female | Control | 55 |
| 155 | Caucasian | Male | Control | 40 |
| 209 | Caucasian | Female | Control | 69 |

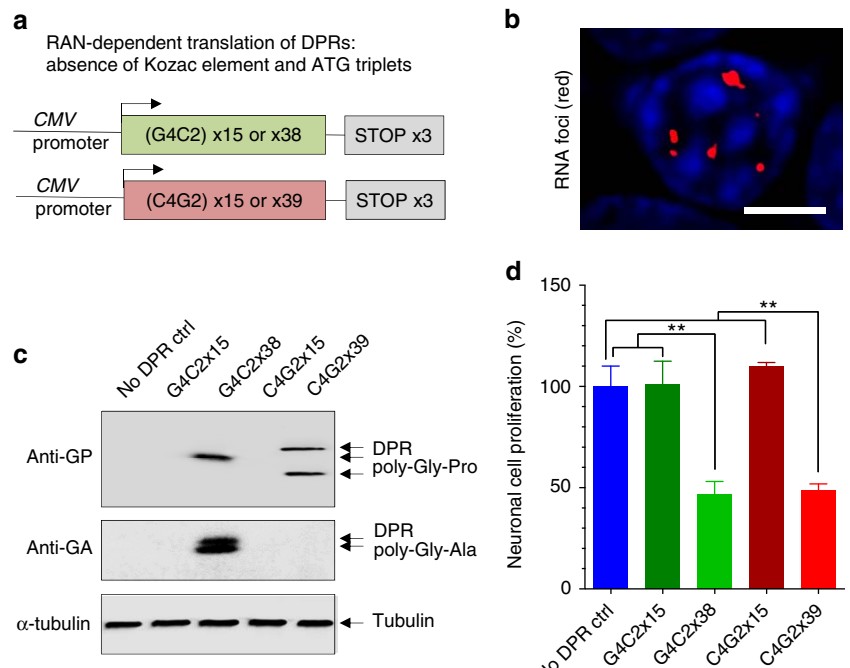

**Figure 4 | Generation of neuronal cell models recapitulating the RAN-dependent translation of sense and antisense DPRs.** (**a**) Diagrammatic representations of constructs. (**b**) Sense G4C2 RNA foci stained with Cy3-labelled antisense C4G2 probe. DAPI was used to stain nuclei of neuronal N2A cells in blue. Scale bar, 5 μm. (**c**) Western blots from N2A cells transfected with a control backbone plasmid (no DPR Ctrl) or the same plasmid expressing either 15 uninterrupted G4C2-sense repeat (G4C2x15), 38 uninterrupted G4C2-sense-repeats (G4C2x38), 15 uninterrupted C4G2-antisense repeat (C4G2x15) or 39 uninterrupted C4G2-antisense-repeats (C4G2x39). Membranes were probed with antibodies against poly-Gly-Pro DPRs, poly-Gly-Ala DPRs or loading control α-tubulin. (**d**) MTT cell proliferation assay performed on N2A cells transfected with either a control backbone plasmid (no DPR Ctrl) or the same plasmid expressing various length of uninterrupted hexanucleotide repeat transcripts in three biological replicate experiments (mean ± s.e.m.; one-way ANOVA with Tukey's correction for multiple comparisons; N (OD650 values) = 12). Statistical significance of data is indicated as follows: NS: non-significant, $P \geq 0.05$; *$P < 0.05$; **$P < 0.01$; ***$P < 0.001$; ****$P < 0.0001$.

repeats (G4C2x15 or C4G2x39) with 3′-end stop codons in each of the three frames (Fig. 4a). The lengths of repeats were confirmed by sequencing and poly-acrylamide gel electrophoresis. The nucleotide sequences are presented in Supplementary Fig. 4e. In mammals, the bulk nuclear export of mRNA is predominantly

coupled to the recruitment of the TREX complex during splicing[53]. Three C9ORF72 transcripts, each containing 4 or 10 introns, are transcribed from the C9ORF72 gene. The synthetic repeat constructs were engineered without splicing elements or intronic sequences to investigate the nuclear export potential of

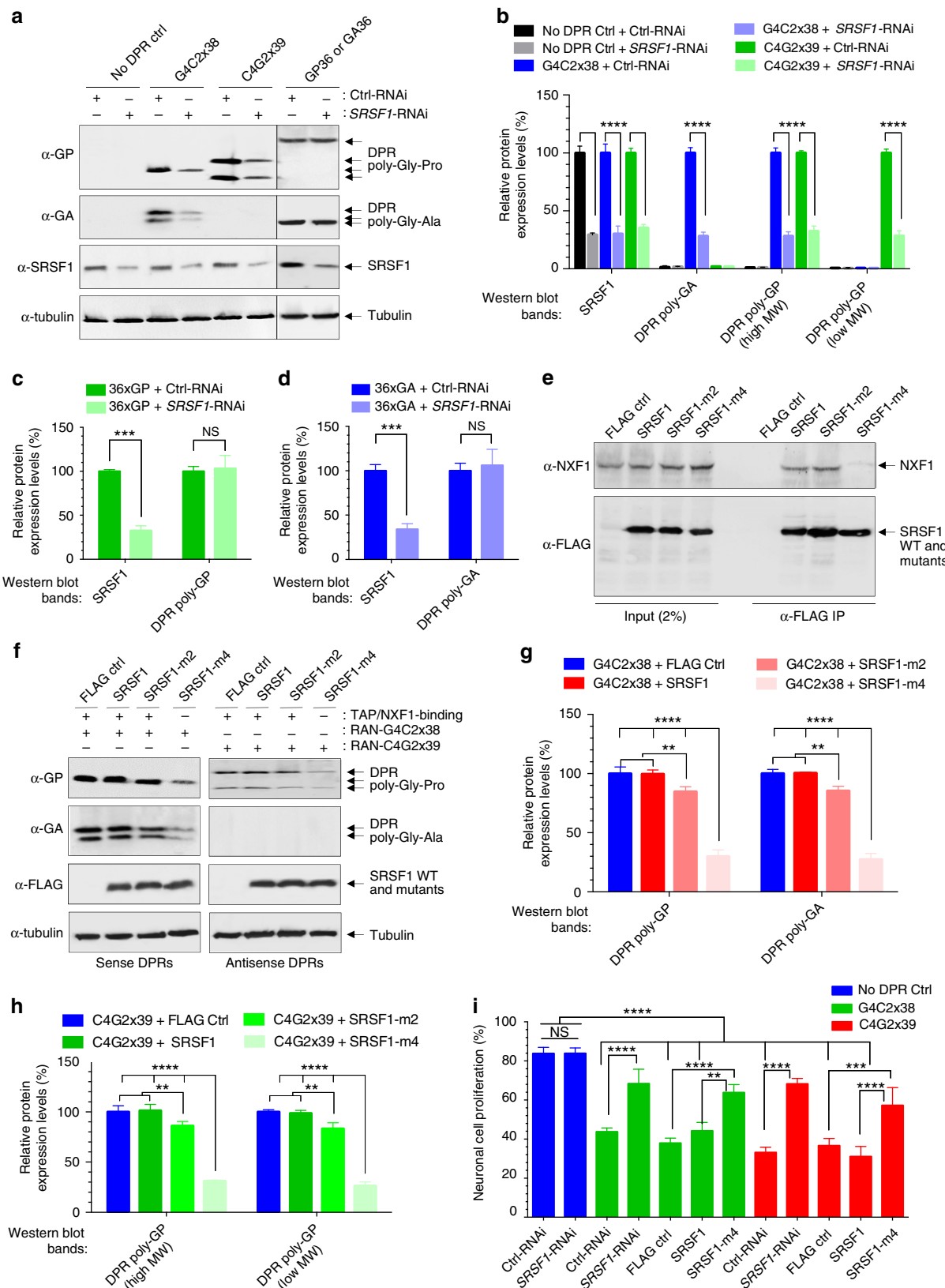

G4C2 and C4G2 repeat RNA sequences in repeat transcripts independently of functional coupling to pre-mRNA splicing. Transfections of mouse neuronal N2A cells with sense or antisense repeat constructs led to the formation of RNA foci for all repeat transcripts (Fig. 4b, data shown for the G4C2x38 construct) and to specific DPR production of poly-Gly-Pro (expressed from both sense and antisense transcripts) and poly-Gly-Ala (expressed from sense transcripts) for repeat transcripts bearing 38 sense or 39 antisense repeats (Fig. 4c). Interestingly, the expression of DPRs correlated with neurotoxic effects in MTT cell proliferation assays, while the control plasmid or the constructs expressing 15 sense or antisense repeats but no DPRs did not exhibit significant cytotoxicity in the neuronal cells (Fig. 4d). We conclude that the minimal G4C2x38 sense and C4G2x39 antisense repeat RNA transcripts can be exported into the cytoplasm independently of functional coupling to pre-mRNA splicing and are substrates for subsequent RAN translation of DPRs.

**SRSF1 depletion inhibits DPRs production in neuronal cells**. A mouse *SRSF1*-knockdown plasmid co-expressing a GFP reporter and a pre-miRNA cassette was engineered similarly to the previously described human *SRSF1*-RNAi (Supplementary Fig. 1b). Transfection of mouse neuronal N2A cells with G4C2x38 or C4G2x39 repeat constructs and the mouse *SRSF1*-RNAi plasmid led to a marked reduction in the RAN translation of both sense and antisense DPRs (Fig. 5a, left panels and Fig. 5b for quantification). The *SRSF1*-RNAi dependent inhibition of DPR production does not depend on the splicing activity of SRSF1 since the RAN constructs are devoid of splicing sites. Moreover, the SRSF1 depletion is specific to the hexanucleotide-repeat sequences since expression of synthetic 36-repeat poly-Gly-Pro (GP36) or 36-repeat poly-Gly-Ala (GA36) DPRs using alternative codons (sequences available in Supplementary Fig. 5) is not altered upon SRSF1 depletion (Fig. 5a, right panels and Fig. 5c–d for respective quantifications of GP36 and GA36). These results support our previous findings that the depletion of SRSF1 did not ameliorate the rough eye phenotypes (Fig. 2f) or the locomotor deficits (Fig. 2g) conferred by G4C2-independent GR36 and PR36 DPRs expression in *Drosophila*.

SRSF1 mediates mRNA nuclear export through binding to NXF1 (refs 34,54). We previously showed that four arginine residues lying in the unstructured linker between the two RNA recognition motifs of SRSF1 (amino acids 11–196) are required both for RNA nuclear export and interaction with NXF1, while

mutations of only two arginine residues lead to slightly reduced binding to NXF1 in human embryonic kidney cells[36]. Similarly, endogenous NXF1 is specifically immunoprecipitated in neuronal N2A cells transfected with FLAG-tagged SRSF1 11-196 wild-type or double R117,118A mutant (SRSF1-m2). In contrast, the co-immunoprecipitation of NXF1 is severely impaired by the quadruple R93,94,117,118A mutations of SRSF1 (SRSF1-m4) (Fig. 5e). Co-transfection of the quadruple SRSF1-m4 dominant mutant further led to a marked reduction in the production of both sense and antisense DPRs while the wild-type sequence or the variant bearing two arginine mutations (SRSF1-m2) respectively had no or little effect (Fig. 5f). This was statistically assessed for both poly-GP and poly-GA DPRs produced by sense repeat transcripts (Fig. 5g) and poly-GP DPRs generated from antisense repeat transcripts (Fig. 5h). Taken together our data demonstrate that the expression of the *C9ORF72* repeat transcripts is dependent on a mechanism that requires the interaction of SRSF1 with the nuclear export receptor NXF1. Accordingly, both the depletion of SRSF1 and the expression of the dominant negative mutant SRSF1-m4 suppress the neurotoxicity mediated by expression of the *C9ORF72* repeat transcripts in neuronal N2A cells (Fig. 5i). Supporting this, RNAi-mediated depletion of the *Drosophila* NXF1 homologue, sbr, could rescue the locomotor deficits in larvae and adult flies expressing G4C2x36 (Supplementary Fig. 6).

**Repeat-sequestration of SRSF1 triggers RNA nuclear export**. Our result showing that expression of the SRSF1-m4 mutant protein acts as a dominant negative mutant for DPR production suggests that the SRSF1-m4 protein is sequestered onto the hexanucleotide repeat transcripts instead of the endogenous SRSF1 protein, preventing in turn interactions of repeat transcripts with NXF1 and nuclear export. Using *in vitro* UV cross-linking assays, we confirmed that the purified recombinant hexa-histidine-tagged SRSF1-m4 protein retains the ability to directly interact with synthetic 5′-end [32]P-radiolabelled sense G4C2x5 (Fig. 6a) and antisense C4G2x5 (Fig. 6b) repeat RNA. These interactions are specific since no binding of RNA was detected in absence of UV–irradiation or with the negative control protein MAGOH. We next sought to investigate whether this was also true in live N2A cells using RNA immunoprecipitation (RIP) assays. N2A cells were transfected with FLAG control, FLAG-tagged SRSF1 or FLAG-tagged SRSF1-m4 and various lengths of sense or antisense repeat transcript constructs prior to fixing of ribonucleoprotein complexes. Cell extracts were then subjected to anti-FLAG

**Figure 5 | SRSF1 depletion and inhibition of the SRSF1:NXF1 interaction inhibit the production of DPRs in C9ORF72-ALS cell models.** (**a**) Western blots from N2A cells co-transfected with either a Ctrl or SRSF1-RNAi vector and control backbone plasmid (no DPR Ctrl) or the same plasmid expressing 38 uninterrupted G4C2-sense-repeats (G4C2x38) or 39 uninterrupted C4G2-antisense-repeats (C4G2x39). Sense/antisense poly-Gly-Pro and sense poly-Gly-Ala DPR proteins are produced by internal repeat RAN translation in the absence of an initiating start codon (nucleotide sequences in Supplementary Fig. 4). A hexanucleotide-repeat specificity control was provided by co-transfection of plasmids expressing poly-Gly-Ala (GA36) or poly-Gly-Pro (GP36) independently of the G4C2/C4G2-repeats (nucleotide sequences in Supplementary Fig. 5) and either Ctrl or SRSF1-RNAi vectors. (**b**) Western blots shown in **a** for the no DPR Ctrl, G4C2x38 and C4G2x39 experiments were quantified in three biological replicate experiments (mean ± s.e.m.; two-way ANOVA with Tukey's correction for multiple comparisons; $N = 3$). (**c,d**) Western blots shown in **a** for the GP36 (**c**) and GA36 (**d**) panels were quantified in three biological replicate experiments (mean ± s.e.m.; two-way ANOVA with Tukey's correction for multiple comparisons; $N = 3$). (**e**) Total extracts from N2A cells transfected with either FLAG control (FLAG ctrl) and either FLAG-tagged SRSF1 aa11-196 wild-type (SRSF1), SRSF1-m2 or SRSF1-m4 are subjected to anti-FLAG immunoprecipitation. Co-immunoprecipitation of endogenous NXF1 is assessed using α-NXF1 antibodies. (**f**) Western blots from N2A cells co-transfected with either G4C2x38 or C4G2x39 plasmids and control (FLAG Ctrl) or FLAG-tagged SRSF1 aa11-196 wild-type (SRSF1), SRSF1-m2 or SRSF1-m4. (**g,h**) Western blots shown in **f** for the G4C2-sense (**g**) and C4G2-antisense (**h**) repeat panels were quantified in three biological replicate experiments (mean ± s.e.m.; two-way ANOVA with Tukey's correction for multiple comparisons; $N = 3$). (**i**) MTT cell proliferation assay performed on N2A cells transfected with either G4C2x38 or C4G2x39 plasmids and Ctrl-RNAi, SRSF1-RNAi, FLAG Ctrl, SRSF1 or SRSF1-m4 in three biological replicate experiments (mean ± s.e.m.; one-way ANOVA with Tukey's correction for multiple comparisons; $N$ (OD650 values) = 12). Statistical significance of data is indicated as follows: NS: non-significant, $P \geq 0.05$; *$P < 0.05$; **$P < 0.01$; ***$P < 0.001$; ****$P < 0.0001$.

immunoprecipitation under the same conditions used in the co-immunoprecipitation of NXF1 (Fig. 5e) prior to the qRT–PCR analysis of SRSF1-crosslinked RNA molecules. Validating our RIP assay, immunoprecipitation of both SRSF1 or SRSF1-m4 led to specific co-precipitation of *SMN*, a known SRSF1-binder[55],

but not of *JUN*, an intronless control transcript not expected to be bound by SRSF1 (ref. 56), in N2A cells expressing either sense (Fig. 6c) or antisense (Fig. 6d) repeat transcripts. In sharp contrast, the levels of immunoprecipitated G4C2-sense or C4G2-antisense repeat transcripts significantly increased with

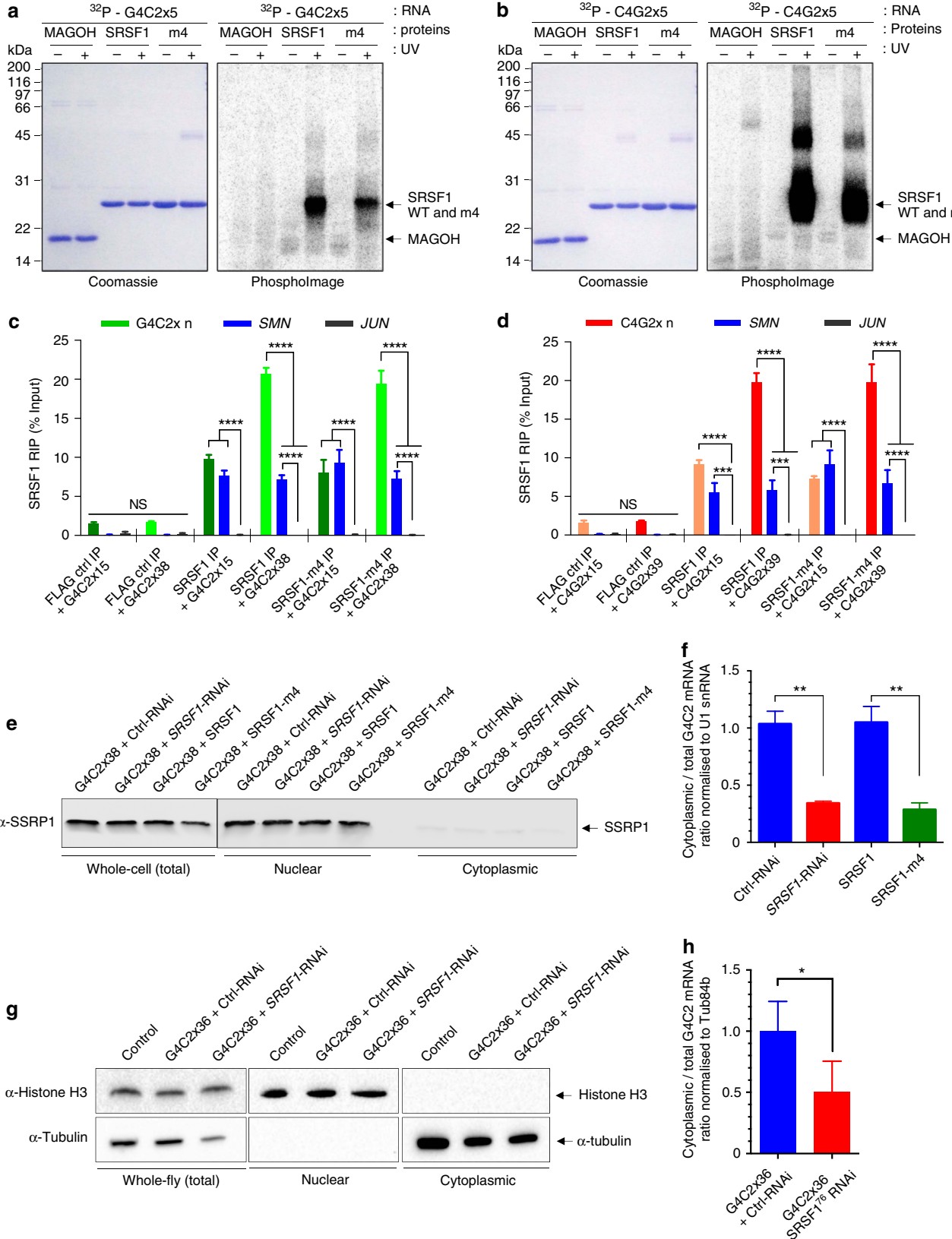

the number of hexanucleotide repeats (Fig. 6c,d) showing length-dependent repeat-RNA sequestration of SRSF1 and SRSF1-m4 in neuronal cells.

To evaluate the effects of SRSF1 depletion or SRSF1-m4 expression on the nuclear export of C9ORF72 repeat transcripts, we measured the total and cytoplasmic levels of G4C2x38 transcripts in the presence of Ctrl or SRSF1-RNAi and SRSF1 or SRSF1-m4 in transfected N2A cells. The quality of the cellular fractionation was checked by immunoblotting using antibodies against the chromatin-remodelling SSRP1 factor (Fig. 6e) showing absence of significant nuclear contamination in the cytoplasmic fractions. Total levels of G4C2x38 transcripts are not significantly altered upon SRSF1-RNAi or expression of SRSF1-m4 while the cytoplasmic levels of G4C2x38 transcripts are markedly reduced in both conditions (Supplementary Fig. 7a). Cytoplasmic repeat transcript levels were also normalized to total levels to specifically assess the nuclear export process as in our previous studies[39,44]. The cytoplasmic/total mRNA level ratios are markedly reduced upon exposure to SRSF1-RNAi (Fig. 6f, Ctrl-RNAi versus SRSF1-RNAi). Similarly, the co-transfection of the SRSF1-m4 mutant which fails to interact with NXF1 led to marked reduction in the normalized cytoplasmic repeat transcript levels (Fig. 6f, SRSF1 versus SRSF1-m4). To extend this analysis in vivo, Drosophila larvae expressing G4C2x36 and either Ctrl or SRSF1-RNAi were subjected to the same fractionation (Fig. 6g) and transcript analysis. As with cells, total levels of G4C2x36 transcripts are not significantly altered upon SRSF1-RNAi while the cytoplasmic levels of G4C2x36 transcripts are markedly reduced (Supplementary Fig. 7b). Consequently, the cytoplasmic/total mRNA level ratio is significantly reduced upon exposure to SRSF1-RNAi (Fig. 6h). Together these data demonstrate that depleting SRSF1 or preventing its repeat RNA-sequestration and interaction with NXF1 specifically inhibit the nuclear export of hexanucleotide repeat transcripts in vitro and in vivo.

**Antagonizing SRSF1 alters DPRs production in primary neurons.** We next sought to validate our findings in primary neurons. Due to high background staining obtained with poly-GP and poly-GA antibodies, we expressed V5 tags in all three frames downstream of the G4C2x38 repeat sequence (Supplementary Fig. 8) to simultaneously detect all DPR species using the more specific and sensitive anti-V5 antibody. Cultured rat cortical neurons were transfected with the G4C2x38-3xV5 construct and either the Ctrl-RNAi, SRSF1-RNAi, SRSF1 or SRSF1-m4 expression plasmids prior to immunofluorescence studies. The nucleotide sequence targeted by the SRSF1-RNAi miRNA hairpin-1 is

identical in human, mouse and rat SRSF1 (Methods, Supplementary Fig. 1b). Microscopy image examples of DPR-negative and DPR-positive neurons are presented in Fig. 7a. The proportion of DPR-positive neurons in approximately 100 successfully transfected neurons from two independent experiments was quantified in each group and all counts were performed blinded. Depletion of SRSF1 led to a significant reduction (25%) in the proportion of neurons with RAN-translated DPR-staining compared to neurons transfected with Ctrl-RNAi (Fig. 7b). Inhibiting the sequestration of endogenous SRSF1 and the interaction with NXF1 by recruitment of the SRSF1-m4 dominant mutant also led to a similar and significant reduction of neurons expressing DPRs compared to neurons transfected with wild-type SRSF1 (Fig. 7c). Only cells showing absence of DPRs were counted DPR-negative. It is very likely that the effects of the SRSF1-RNAi or SRSF1-m4 expression have been under-estimated since neurons expressing reduced amounts of DPRs would still have been scored as DPR-positive. We concluded that our findings in N2A cells were corroborated in cultures of primary neurons.

**SRSF1 depletion targets human C9ORF72 repeat transcripts.** In order to investigate the nuclear export of C9ORF72 transcripts in the context of wild-type and repeat-expanded C9ORF72 genes, we differentiated motor neurons from established induced-neural progenitor cells (iNPCs) derived from sex/age matched control and C9ORF72-ALS patient fibroblasts[52]. Both control and C9ORF72-ALS induced iNeurons express the neuronal lineage marker Tuj1 and exhibit the propensity to form complex branching (Supplementary Fig. 9). High content imaging analysis of axonal length (Fig. 8a) and soma cell size (Fig. 8b) did not show any significant differences between control and C9ORF72-ALS differentiated iNeurons under basal culture conditions, in agreement with previous reports[9,10]. To test a potential neuroprotective effect of SRSF1 depletion in disease relevant cells, we next differentiated iNPCs into motor neurons (iMNs), transduced them with an adenoviral vector expressing RFP under the Hb9 promoter and cultured them either in monoculture or in co-culture with control or ALS-derived iAstrocytes. We did not observe increased cell death or morphological abnormalities when the C9ORF72-ALS iMNs were cultured without astrocytes (data not shown). Remarkably, however, the transduction of SRSF1-RNAi lentivirus in iMNs prior to co-cultures with C9ORF72-ALS iAstrocytes resulted in significantly higher survival of iMNs against the astrocytic-derived toxicity (Fig. 8c) indicating a neuroprotective effect of

**Figure 6 | Depleting SRSF1 or inhibiting its repeat-RNA sequestration inhibit the nuclear export of C9ORF72 RAN-translated transcripts.**
(**a,b**) Protein:RNA UV crosslinking assays using purified recombinant proteins and 32P-end-radiolabelled G4C2x5 (**a**) and C4G2x5 (**b**) repeat RNA probes. Proteins are visualized on SDS–PAGE stained with Coomassie blue (left panels) and covalently linked RNA:protein complexes by autoradiography on PhosphoImages (right panels). (**c,d**) RNA immunoprecipitation (RIP) assays. Formaldehyde was added to the medium of live N2A cells co-transfected with G4C2x15, G4C2x38 (**c**), C4G2x15 or C4G2x39 (**d**) and either FLAG control (FLAG Ctrl), FLAG-tagged SRSF1 aa11-196 wild-type (SRSF1) or SRSF1-m4 were subjected to anti-FLAG immunoprecipitation. Purified RNA was analysed by qRT–PCR following normalization to U1 snRNA levels in three biological replicate experiments (mean ± s.e.m.; two-way ANOVA with Tukey's correction for multiple comparisons; N (qRT–PCR reactions) = 6). (**e**) Western blots of N2A cells co-transfected with G4C2x38 and either Ctrl or SRSF1-RNAi plasmids or with G4C2x38 and either FLAG-tagged SRSF1 aa11-196 wild-type (SRSF1) or SRSF1-m4, subjected to cellular fractionation using hypotonic lysis to yield cytoplasmic fractions. The chromatin remodelling SSRP1 factor is used to check for potential nuclear contamination. (**f**) Cytoplasmic and total G4C2-repeat sense transcript levels were normalized to U1 snRNA levels in three biological replicate experiments prior to plotting as a ratio to account for potential changes in mRNA transcription/stability (mean ± s.e.m.; one-way ANOVA with Tukey's correction for multiple comparisons; N (qRT–PCR reactions) = 6). (**g**) Western blots of Drosophila expressing G4C2x36 and either Ctrl or SRSF1-RNAi, subjected to cellular fractionation using hypotonic lysis to yield cytoplasmic fractions. Histone H3 is used to check for potential nuclear contamination. (**h**) Cytoplasmic and total G4C2-repeat sense transcript levels were normalized to Tub84b levels in three biological replicate experiments prior to plotting as a ratio to account for potential changes in mRNA transcription/stability (mean ± s.e.m.; paired two-tailed t-test; N (qRT–PCR reactions) = 3). In contrast to cytoplasmic levels, total levels of hexanucleotide repeat transcripts were not significantly altered upon expression of SRSF1-m4 or depletion of SRSF1 in cells or flies (Supplementary Fig. 7). Statistical significance of data is indicated as follows: NS: non-significant, $P \geq 0.05$; *$P < 0.05$; **$P < 0.01$; ***$P < 0.001$; ****$P < 0.0001$.

SRSF1 depletion in motor neurons derived from C9ORF72-ALS patients. Consistent with our previous data presented in *Drosophila* and neuronal cells models, we also found that depletion of SRSF1 in C9ORF72-ALS patient-derived iMNs leads to specific reduction in the expression levels of poly-GP DPRs although it appears less efficient in C9ORF72-ALS patient 78 (Fig. 8d).

We next quantified the total, nuclear and cytoplasmic levels of intron-1-spliced *C9ORF72* transcripts (using qPCR primers annealing in exon-1 and exon-3)[27,28] and unspliced *C9ORF72* transcripts retaining intron-1 (using qPCR primers annealing in exon-1 and in intron-1 upstream of the hexanucleotide repeat expansion)[27,28] to evaluate the potential impact of *SRSF1*-RNAi on the splicing of intron-1 and on the nuclear export of wild-type and pathological *C9ORF72* transcripts. Depletion of the chromatin-remodelling factor SSRP1 in the cytoplasmic fractions and of actin in the nuclear fractions was used to validate the quality of the subcellular fractionation (Fig. 8e). The relative expression levels of *SRSF1* mRNA were down regulated by approximately 80% upon *SRSF1*-RNAi transduction of iNeurons

differentiated from either two control or two sex/age-matched C9ORF72-ALS patient lines (Supplementary Fig. 10). No significant changes in the total, nuclear or cytoplasmic levels of intron-1-spliced transcripts were measured between control and C9ORF72-ALS iNeurons transduced or not with *SRSF1*-RNAi lentivirus (Fig. 8f). The mRNA level ratios of *SRSF1*-RNAi (MOI5) over untreated (MOI0) were further plotted to assess the net impact of the *SRSF1*-RNAi on each cellular compartment and control or C9ORF72-ALS iNeurons (Fig. 8g). These data show that the proportion of exon1-exon3 spliced transcripts is not altered in C9ORF72-ALS neurons and that the presence of the hexanucleotide repeat expansion does not affect the splicing of intron-1, in full agreement with a recent study[27]. In addition, we also show here that the *SRSF1*-RNAi has no effect on the nuclear export or the splicing of intron-1 in the spliced transcripts that lead to the production of the wild-type C9ORF72 protein. The same experimental analysis was carried out for the *C9ORF72* transcripts retaining intron-1 (Fig. 8h,i). We were unable to detect significant levels (above non template control qRT–PCR reactions) of intron-1-retaining *C9ORF72* transcripts in the

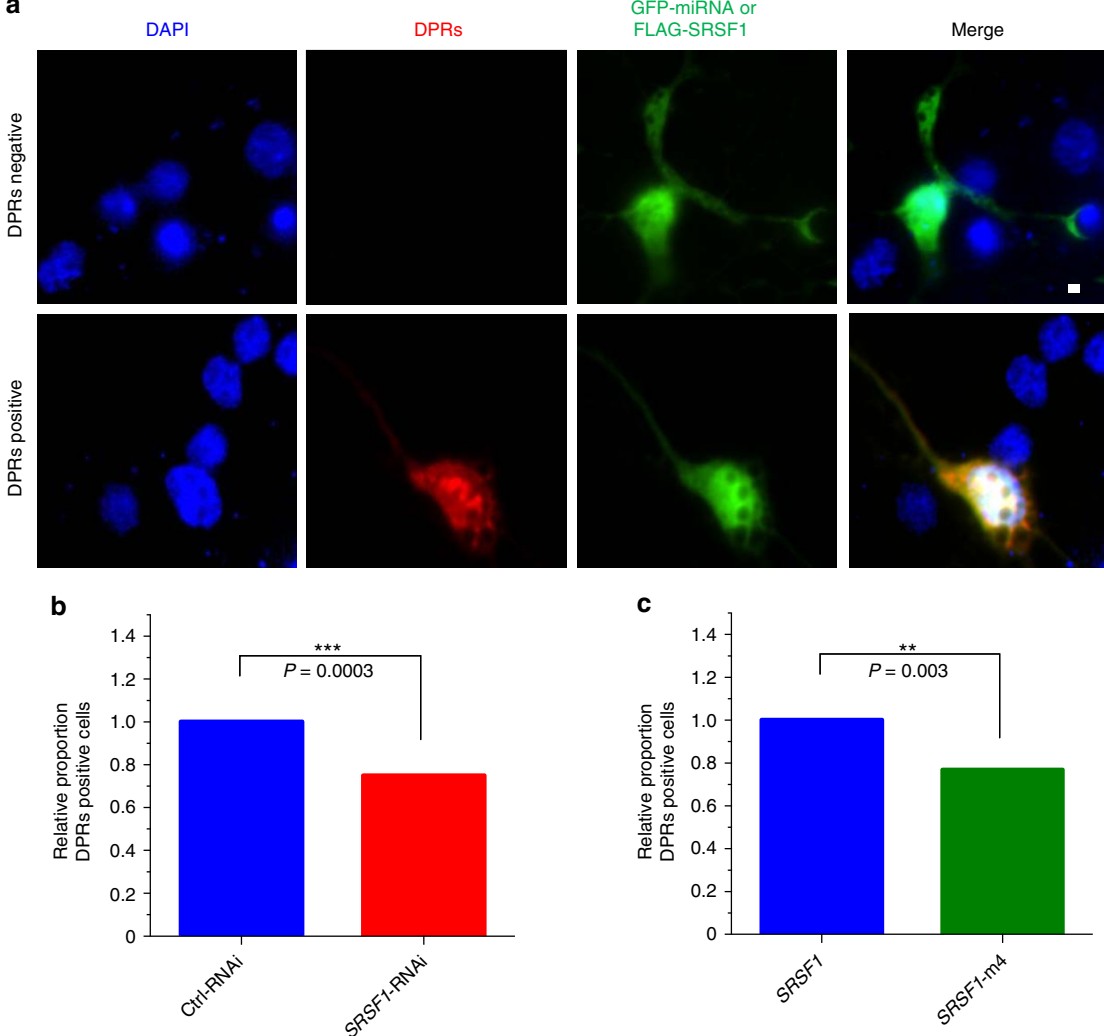

**Figure 7 | Depleting SRSF1 or inhibiting its repeat-RNA sequestration and interaction with NXF1 inhibit the production of DPRs in primary neurons.** (**a**) Immunofluorescence microscopy of cultured rat cortical neurons. DPRs were detected in the red channel using anti-V5 and anti-mouse ALEXA594 antibodies. The Ctrl-RNAi and SRSF1-RNAi constructs co-express GFP while the FLAG-tagged SRSF1 proteins were stained using an anti-FLAG antibody conjugated to FITC allowing detection and quantification of transfected neurons in the green channel. Scale bar: 5 μm. (**b,c**) Statistical assessment of the cortical neuron counts was performed from approximately 100 transfected neurons for each group (Fisher's exact test; *N* (transfected neurons) = Ctrl-RNAi: 95, SRSF1-RNAi: 112, SRSF1: 106, SRSF1-m4: 121).

cytoplasm of control iNeurons consistent with the nuclear retention of unspliced transcripts. In striking contrast, the presence of the hexanucleotide repeat expansion in C9ORF72-ALS patients triggers the nuclear export of C9ORF72 repeat transcripts retaining intron-1 (Fig. 8h) consistent with our previous data showing that the sequestration of SRSF1 on

synthetic hexanucleotide repeat expansions promotes nuclear mRNA export through the interaction with NXF1 (Fig. 6). Whilst the depletion of *SRSF1* did not affect the total level and biogenesis/stability of intron-1-retaining transcripts in control or C9ORF72-ALS iNeurons, it specifically triggers a cytoplasmic reduction and nuclear accumulation of pathological *C9ORF72*

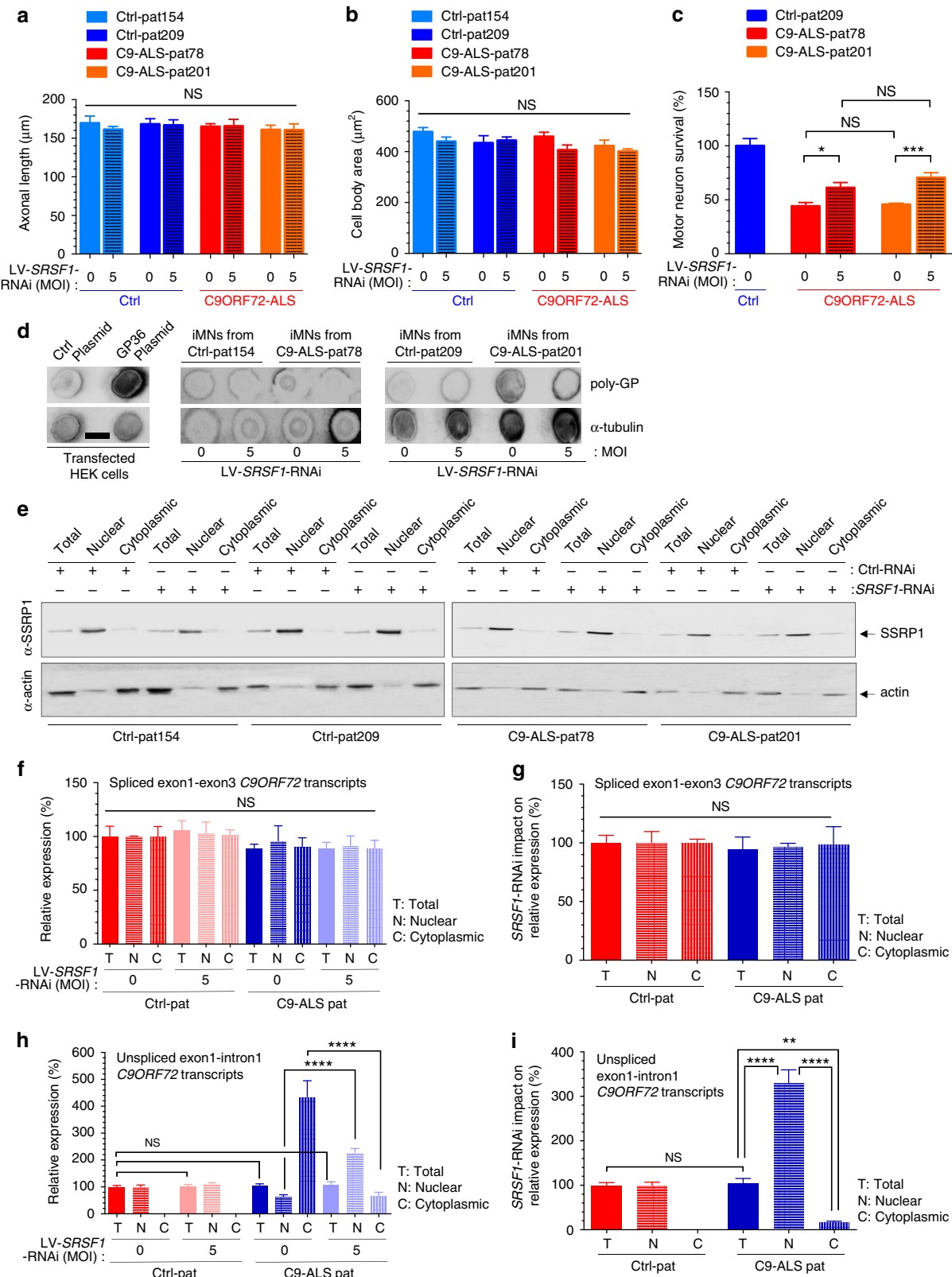

repeat transcripts in C9ORF72-ALS iNeurons (Fig. 8h,i). These data demonstrate that the depletion of SRSF1 specifically inhibits the nuclear export of expanded *C9ORF72* repeat transcripts. Taken together, our data show that the SRSF1 depletion has no effect on the expression level, splicing or nuclear export of wild-type spliced exon1-exon3 C9ORF72 transcripts while it specifically inhibits the nuclear export of pathological *C9ORF72* transcripts retaining the hexanucleotide repeats in intron-1.

## Discussion

Microsatellite expansions of 3–6 nucleotides in coding and non-coding regions of genes cause neurodegeneration through complex mechanisms involving protein loss-of-function and protein/RNA toxic gain-of-function mechanisms[57]. The production of toxic polymeric repeat proteins by RAN translation has now been characterized in multiple neuro-degenerative disorders caused by microsatellite expansions including spinocerebellar ataxia type 8 (SCA8)[58], myotonic dystrophy type 1 (DM1)[58], Fragile X-associated tremor and ataxia syndrome (FXTAS)[59], C9ORF72-ALS[6,13–16,27] and Huntington disease (HD)[60]. However, the mechanisms involved in the nuclear export of these disease-related repeat transcripts are currently unknown.

We previously suggested that the sequestration of nuclear export adaptors onto *C9ORF72* repeat transcripts might trigger the abnormal nuclear export of *C9ORF72* repeat transcripts and the subsequent RAN translation of DPRs in the cytoplasm[12]. In this study, we identified for the first time the molecular mechanism driving the nuclear export of pathological *C9ORF72* repeat transcripts. We investigated whether the partial depletion of two evolutionarily conserved nuclear export adaptors which avidly interact with the hexanucleotide repeat transcripts[12], ALYREF and SRSF1, would mitigate DPR-mediated neurotoxicity in an established *Drosophila* model of C9ORF72-ALS[16]. We discovered that the partial depletion of SRSF1 prevents *in vivo* neurodegeneration and suppresses the associated locomotor deficits while the depletion of ALYREF only had marginal effects. The depletion of SRSF1 in

C9ORF72-ALS patient-derived motor neurons also conferred neuroprotection of motor neurons in co-culture with C9ORF72-ALS astrocytes. Moreover, we also showed that this intervention does not affect the morphology or the growth of control and C9ORF72-ALS patient-derived motor neurons. On the other hand, the depletion of *SRSF1* in patient-derived C9ORF72-ALS astrocytes significantly suppressed motor neuron death in a co-culture system. The mechanisms for suppression of astrocyte-mediated neurotoxicity remain however to be determined. They might involve a modification of the RNA or protein composition in the extra-cellular exosomes released by astrocytes.

Using neuronal N2A cells, we demonstrated that the nuclear export of *C9ORF72* repeat transcripts and subsequent RAN translation depends on the interaction of SRSF1 with the nuclear export receptor NXF1. Depleting SRSF1 or inhibiting its endogenous RNA-repeat sequestration and interaction with NXF1 lead to a marked inhibition of the nuclear export of *C9ORF72* repeat transcripts and RAN translation of sense and antisense DPRs to prevent C9ORF72 repeat-mediated neurotoxicity. We also showed that the SRSF1-dependent inhibition of the nuclear export of *C9ORF72* repeat transcripts leads to altered production of DPRs in *Drosophila* and patient-derived motor neurons. We noted some variability in the efficiency of the SRSF1 depletion in patient-derived motor neurons. This might be due to variation between expression levels of DPRs and/or genetic background of patients-derived cells. Importantly, the depletion of SRSF1 in control or C9ORF72-ALS patient-derived neurons does not affect the expression levels or the nuclear export of intron-1-spliced transcripts required for the translation of the wild-type C9ORF72 protein. This also indicates that the nuclear export of non-repeat *C9ORF72* transcripts does not depend on the nuclear export adaptor SRSF1. In sharp contrast to control neurons, the presence of the hexanucleotide repeat expansion in intron-1 of *C9ORF72* transcripts led to SRSF1-dependent mRNA nuclear export, while depletion of SRSF1 specifically inhibits the nuclear export but not the levels or splicing of *C9ORF72* transcripts retaining expanded hexanucleotide repeats in intron-1. Taking

**Figure 8 | Depletion of SRSF1 specifically inhibits the nuclear export of pathological C9ORF72 repeat transcripts in patient-derived neurons.** (**a**) The axonal length of patient-derived iMNs treated or not with SRSF1-RNAi was assessed by high content imaging in three biological replicate experiments (mean ± s.e.m.; one-way ANOVA with Tukey's correction for multiple comparisons, NS: non-significant; $N$ (average axon length/well) = 9). (**b**) The cell body area of patient-derived iMNs treated or not with SRSF1-RNAi was assessed by high content imaging in three biological replicate experiments (mean ± s.e.m.; one-way ANOVA with Tukey's correction for multiple comparisons; $N$ (average cell body area/well) = 9). (**c**) The survival of patient-derived Ctrl or C9ORF72-ALS iMNs treated or not with SRSF1-RNAi was quantified in co-cultures with patient-derived Ctrl or C9ORF72-ALS iAstrocytes in six biological replicate experiments at day 4 (mean ± s.e.m.; one-way ANOVA with Tukey's correction for multiple comparisons; $N$ (iMNs) = Ctrl-pat209: 142/165/174/117/122/168, C9-ALS-pat78: 77/65/41/68/71/70; C9-ALS-pat78 + SRSF1-RNAi: 106/84/81/97/113/83; C9-ALS-pat201: 68/69/68/62/66/74; C9-ALS-pat201 + SRSF1-RNAi: 131/104/96/82/111/104). (**d**) Total protein extracts from HEK cells transfected with either Ctrl or GP36 plasmids and patient-derived iMNs transduced (MOI = 5) or not (MOI = 0) with LV-SRSF1-RNAi viruses are analysed by dot blots using poly-GP DPRs and loading control α-tubulin antibodies. Scale bar: 6 mm. (**e**) Western blots of iNeurons differentiated from control (Ctrl-pat 154, Ctrl-pat155) and C9ORF72-ALS (C9-ALS-pat78, C9-ALS-pat183) patients treated or not with LV-SRSF1-RNAi (MOI 0 or 5) were subjected to cellular fractionation using hypotonic lysis to yield cytoplasmic fractions. The chromatin remodelling SSRP1 factor is used to check for potential nuclear contamination in cytoplasmic fractions. Depletion of actin in nuclear fractions was used to check for quality of the nuclear fractions. The efficacy of SRSF1-RNAi was also validated by qRT–PCR achieving ~80% SRSF1 mRNA knockdown (Supplementary Fig. 10). (**f**) Total, nuclear and cytoplasmic levels of intron1-spliced C9ORF72 transcripts (as measured by the exon1-exon3 junction) were quantified in two technical replicates for each of two biological replicate experiments by qRT–PCR following normalization to U1 snRNA levels and to 100% in control patients at MOI0 (mean ± s.e.m.; one-way ANOVA with Tukey's correction for multiple comparisons, NS: non-significant; $N$ (qRT–PCR reactions) = 8). (**g**) Intron1-spliced C9ORF72 transcripts levels normalized to U1 snRNA levels (**d**) were plotted as a ratio SRSF1-RNAi MOI5 over MOI0 to evaluate the specific effect due to the SRSF1-RNAi (mean ± s.e.m.; one-way ANOVA with Tukey's correction for multiple comparisons; $N$ (qRT–PCR reactions) = 8). (**h**) Total, nuclear and cytoplasmic levels of unspliced C9ORF72 transcripts retaining intron1 (as measured by the exon1–intron1 junction) were quantified in two technical replicates for each of two biological replicate experiments by qRT–PCR following normalization to U1 snRNA levels and to 100% for control patients at MOI0 (mean ± s.e.m.; one-way ANOVA with Tukey's correction for multiple comparisons, NS: non-significant; $N$ (qRT–PCR reactions) = 8). (**i**) Unspliced C9ORF72 transcripts retaining intron1 levels normalized to U1 snRNA levels (**f**) were plotted as a ratio SRSF1-RNAi MOI5 over MOI0 to evaluate the specific effect due to the SRSF1-RNAi (mean ± s.e.m.; one-way ANOVA with Tukey's correction for multiple comparisons; $N$ (qRT–PCR reactions) = 8). Statistical significance of data is indicated as follows: NS: non-significant, $P \geq 0.05$; *$P < 0.05$; **$P < 0.01$; ***$P < 0.001$; ****$P < 0.0001$.

these data together, we show that sequestration of SRSF1 onto *C9ORF72* hexanucleotide repeats is able to license the NXF1-dependent nuclear export of pathological *C9ORF72* repeat transcripts without functional coupling of the nuclear export process to pre-mRNA splicing. This explains in turn why the depletion of SRSF1 has no effect on the level, intron-1-splicing or nuclear export of wild-type *C9ORF72* transcripts.

In conclusion, we have elucidated for the first time the molecular mechanism driving the nuclear export of pathological *C9ORF72* repeat transcripts which allows for RAN translation of dipeptide repeat proteins in the cytoplasm (Fig. 9a). The depletion of SRSF1 specifically inhibits the nuclear export of the pathologically expanded *C9ORF72* transcripts without interfering with biogenesis/processing of the wild-type *C9ORF72* transcripts (Fig. 9b). The expression of the engineered SRSF1-m4 protein,

which retains specific ability to be sequestered on repeat transcripts but fails to effectively interact with NXF1, also inhibits the nuclear export of repeat transcripts and the production of DPRs. (Fig. 9c). Both these interventions represent promising prospects for the development of an effective neuroprotective strategy in C9ORF72-related ALS. The effects of antagonizing SRSF1 in the vertebrate brain remain to be elucidated in wild-type mice as well as in murine C9ORF72-ALS models. Interestingly, it was recently shown that whilst SRSF1 directly binds thousands of transcripts, the depletion of *SRSF1* in isolation affects the nuclear export of only 225 transcripts (<1% transcribed coding genes) due to the presence of 6 additional SRSF factors (SRSF2-7) which act as redundant NXF1-dependent nuclear export adaptors[46]. The cellular pathways causing C9ORF72 repeat-mediated neurodegeneration and the precise mechanism(s) of

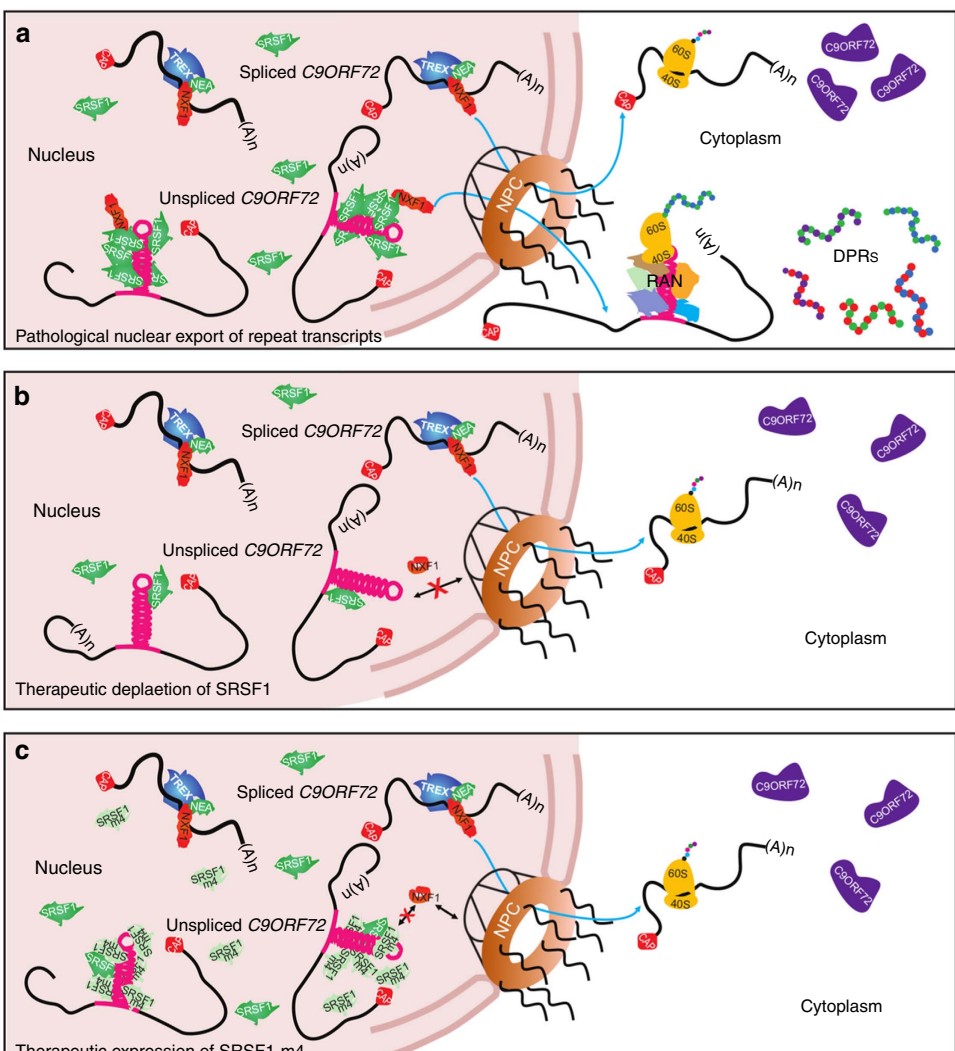

**Figure 9 | Model for the nuclear export of pathological C9ORF72 hexanucleotide repeat transcripts and therapeutic manipulation.** (**a**) The nuclear export of sense and antisense C9ORF72 transcripts retaining expanded hexanucleotide repeats in intron1 specifically depend on the sequestration of SRSF1 and its interaction with the nuclear export receptor NXF1. In contrast, the nuclear export of intron1-spliced C9ORF72 transcripts required for the production of the C9ORF72 protein does not involve the interaction of SRSF1 with NXF1 however the nuclear export adaptor(s) (NEA) remain to be identified. (**b**) The depletion of SRSF1 specifically inhibits the nuclear export of C9ORF72 transcripts retaining expanded hexanucleotide repeats in intron1, likely due to a reduction in the sequestration of endogenous SRSF1 onto the C9ORF72 hexanucleotide repeats and failure to abnormally remodel NXF1 in a high RNA-binding mode, while it does not affect the expression levels and splicing/retention of intron1. Moreover, the depletion of SRSF1 does not affect the expression levels, the splicing of intron1 or the nuclear export of wild-type intron1-spliced C9ORF72 transcripts required for the production of the C9ORF72 protein. (**c**) Over-expression of the SRSF1-m4 protein, which fails to interact efficiently with NXF1, competes endogenous SRSF1 for sequestration onto hexanucleotide repeats preventing in turn interactions with NXF1 and nuclear export of C9ORF72 repeat transcripts.

neuroprotection conferred by the targeting of SRSF1 remain to be elucidated in future studies.

Inhibiting the nuclear export of repeat transcripts might also confer neuroprotection in other microsatellite expansion disorders. However, it will remain essential to determine pathophysiological contributions between polymeric repeat protein production and RNA-mediated toxicity by nuclear retention of transcripts and/or sequestration of RNA-processing factors on repeat transcripts. While expression of repeat proteins can kill cells *in vitro*, it is difficult to evaluate the levels of RAN-translation in patients and the thresholds required for triggering neurotoxicity which will differ depending on the nature of the repeat expansions, the disease in question and the cell types. There is however growing evidence for a pathogenic role of RAN-translation and the data presented here fully support this. For example, FXTAS was initially thought to be caused by intranuclear retention of transcripts and sequestration of splicing factors[61,62]. However, the discovery of RAN translation in the same model challenged this view[59]. Similarly, in C9ORF72-ALS, a 10-fold increase in the number of intranuclear RNA foci does not significantly alter survival or global RNA processing, while expression of DPRs caused neurodegeneration[27] in full agreement with the data presented here. Partial depletion of individual nuclear export adaptors does not appear to be detrimental to the functioning of higher eukaryotic cells. Therefore, they might constitute viable therapeutic targets for inhibiting the nuclear export of repeat transcripts and the production of toxic repeat proteins, particularly in neurodegenerative diseases where RAN-translation appears to have a prominent pathological role.

## Methods

***Drosophila* husbandry and locomotor assays.** *Drosophila* were raised under standard conditions on a molasses, cornmeal and yeast based food in 12 h:12 h light:dark cycle at 25 °C unless otherwise stated. All *C9ORF72* transgenic lines[16] were a gift from Adrian Isaacs and Linda Partridge (University College London). *GMR-GAL4, D42-GAL4, nSyb-GAL4, da-GAL4, UAS-sbr*-RNAi [P{TRiP.HM05135}attP2] and *UAS-luciferase*-RNAi [P{TRiP.JF01355}attP2], used as control-RNAi, were obtained from the Bloomington *Drosophila* Stock Centre (Bloomington, IN). Other UAS-RNAi lines were obtained from the Vienna *Drosophila* Resource Centre. References and sequences of insertions are provided in Supplementary Note 1. For larval crawling assays, transgenes were expressed by *nSyb-GAL4* and animals were grown at 29 °C. Wandering third instar larvae were collected from vials, rinsed with distilled water, and placed in petri dishes with a 1% agarose matrix. Larvae were observed directly for 2 min and the number of peristaltic waves recorded. Climbing assays were performed as described[63] with transgenic expression induced by *D42-GAL4*. Adult flies were tested 1-3 days after eclosion. Male flies were used for all experiments.

***Drosophila* light microscopy imaging and scanning electron microscopy.** Eye phenotypes were analysed by induction of transgene expression by *GMR-GAL4* raised at 25 °C. For light microscopy of *Drosophila* eyes, stacks of images were collected on a Nikon motorized SMZ stereo zoom microscope fitted with 1x Apo lens. Extended focus images were generated using Nikon Elements software. Scanning electron microscopy (SEM) was performed according to a standard protocol[64] and images were captured using a Philips XL-20 SEM microscope. All animals of a given genotype displayed essentially identical phenotypes and randomly selected representative images are shown.

**Plasmids.** FLAG-tagged SRSF1/SF2/ASF plasmids were generated in ref. 36. MicroRNA sequences were designed using the 'miR RNAi' Block-IT RNAi designer tool (ThermoFisher) at Sequences are presented in Supplementary Note 2. Synthesized oligonucleotides (Sigma) were annealed and ligated into pcDNA6.2-GW/EmGFP using the BLOCK-iT PolII-miR-RNAi Expression Vector Kit with EmGFP (ThermoFisher, catalogue number K4936-00). For chaining, BamHI/XhoI-restricted pre-miR2-RNAi cassettes were subcloned into pcDNA6.2-GW/EmGFP-*SRSF1*-miR1 using BglII/XhoI. The PCR fragments encompassing EmGFP and the chained *SRSF1* pre-miRNA cassette were additionally cloned into the lentiviral plasmid SIN-PGK-cPPT-GDNF-WHV(9)[65] using SpeI/XhoI.

Uninterrupted C9ORF72 hexanucleotide sense GGGGCCx15 or GGGGCCx38 and antisense CCCCGGx15 or CCCCGGx39 repeats were built using the synthetic oligonucleotides 5′-(GGGGCC)$_{15}$-3′ and 5′-CCCC-(GGCCCC)$_{14}$-GG-3′. Oligonucleotides were annealed by heating to 99 °C for 30 min and cooling

0.5 °C min$^{-1}$ to ambient with incubation at 70 °C for 10 min. Oligonucleotides were phosphorylated with T4-Polynucleotide Kinase (New England Biolabs), ligated using T4 DNA Ligase (ThermoFisher) and treated with Mung Bean nuclease (New England Biolabs) for blunt ligation. The band corresponding to trimeric oligonucleotides was gel-purified and ligated into pcDNA3.1 (Invitrogen) with blunted EcoRI ends to allow cloning in both sense and antisense orientation. Sequences are presented in Supplementary Fig. 4.

Synthetic sequences encoding poly-Gly-Pro and poly-Gly-Ala x36 DPRs independently of G4C2 repeats were first cloned into pcDNA3.1 using EcoRI/NotI. Synthetic sequences encoding poly-Gly-Pro and poly-Gly-Ala x36 were subcloned using BamHI/NotI into pCI-neo-V5-N using BclI/NotI. BclI restriction site was previously introduced into pCI-neo-V5-N by site directed mutagenesis using forward actctagaggtaccacgtgatcattctcgagggtgctatccaggc and reverse gcctggatagcaccctcgagaatgatcacgtggtacctctagagt primers.

**Lentivirus production.** Twenty 10 cm dishes seeded with $3 \times 10^6$ HEK293T cells/dish were each transfected with 13 μg pCMVΔR8.92, 3.75 μg pM2G, 3 μg pRSV and 13 μg SIN-CMV-miRNA using calcium phosphate transfection[65]. Media was replaced after 12 and 48 h later, supernatant was collected, filtered through a 0.45 μm filter and centrifuged at 19,000 r.p.m. for 90 min at 4 °C using a SW28 rotor (Beckman). The viral pellet was re-suspended in PBS with 1% BSA and stored at −80'C. The biological titre of the virus was determined by transducing HeLa cells with $10^{-2}$, $10^{-3}$ and $10^{-4}$ dilutions of the vector. 72 h post-transduction, the percentage of GFP positive cells was measured with a Fluorescent-Activated cell sorter (FACS, LSRII). The biological titre is expressed as the number of transducing units per ml (TU/ml) and is calculated as follows: Vector titer = [(% positive cells × total number of cells) × dilution factor × 2].

**Tissue culture.** Cells were maintained in a 37 °C incubator with 5% CO$_2$. HEK293T **c**ells were cultured in Dulbecco's Modified Eagle Medium (Sigma) supplemented with 10% fetal bovine serum (FBS) (Gibco) and 5 U ml$^{-1}$ Penstrep (Lonza). Neuro-2a (N2A) (ATCC) cells were cultured in Dulbecco's Modified Eagle Medium (Sigma) supplemented with 10% FBS (Gibco), 5 U ml$^{-1}$ Penstrep (Lonza) and 5 mM sodium pyruvate.

Hb9GFP mouse stem cells were cultured as described[66] and differentiated into motor neurons with 2 μM retinoic acid (Sigma) and 1 μM Smoothened Agonist (SAG) (Millipore) for 5 days. Embryoid bodies were then dissociated with papain and sorted using the FACSAria III (BD Biosciences).

For patient-derived cell cultures, informed consent was obtained from all subjects before sample collection (Study number STH16573, Research Ethics Committee reference [12]/YH/0330). Human patient-derived astrocytes (iAstrocytes) were differentiated from induced Neural Progenitor Cells (iNPCs) as described[52] and cultured in DMEM Glutamax (Gibco) with 10% FBS (Sigma) and 0.02% N2 (Invitrogen). Human patient-derived neurons (iNeurons) were differentiated from the previously established iNPCs using a modified version of the protocol described in reference[52]. Briefly, 30,000 iNPCs were plated in a 6-well plate coated with fibronectin (Millipore) and expanded to 70–80% confluence. At confluence, iNPC medium was replaced with neuron differentiation medium (DMEM/F-12 with Glutamax supplemented with 1%N2, 2%B27 (Gibco)). On day one of differentiation the cells were treated with 2.5 μM of DAPT (Tocris). On day three the neuron differentiation medium is supplemented with 1 μM retinoic acid (Sigma), 0.5 μM Smoothened Agonist (SAG) (Millipore) and 2.5 μM Forskolin (Sigma) for 7 days. To obtain iMotor Neurons, iNeurons were re-plated on fibronectin and cultured in retinoic acid, SAG and Forskolin for 14 more days with addition of BDNF, CNTF and GDNF (all at 20 ng ml$^{-1}$) for the last 10 days of differentiation. For SRSF1 knockdown, cells were transduced with lentivirus expressing control GFP or human *SRSF1*-RNAi co-expressing GFP at an MOI of 5 at day 14 along with the HB9:RFP adenovirus.

**Co-cultures of patient-derived iAstrocytes and motor neurons.** For Hb9GFP motor neuron and patient-derived iAstrocyte co-cultures, 20,000 induced neural progenitor cells (iNPCs) were plated in 6-well plates in astrocyte medium. The day after plating, cells were transduced with lentivirus expressing control GFP or human *SRSF1*-RNAi at an MOI of 5, 7 or 10. The human *SRSF1*-RNAi virus also co-expressed GFP for evaluating transduction efficiency. Co-culture experiments were performed using GFP+ motor neurons from Hb9GFP+ mouse stem cells and i-Astrocytes transduced with an adenovirus expressing red fluorescent protein (Ad-RFP). Seven days post-transduction with Ad-RFP and LV-*SRSF1*-RNAi, iAstrocytes were plated at a density of 10,000 cell per well. The day after, Hb9GFP embryoid bodies were dissociated and sorted for GFP+ cells. 10,000 GFP+ motor neurons were plated onto the astrocytes in motor neuron medium consisting of DMEM/F12, 2% horse serum (Invitrogen), 2% N2, 2% B27 plus GDNF (Invitrogen; 10 ng ml$^{-1}$), BDNF (Invitrogen; 10 ng ml$^{-1}$), CNTF (Invitrogen; 10 ng ml$^{-1}$) and IGF-1 (Invitrogen; 10 ng ml$^{-1}$). Nine 10× images/well to cover the whole well surface were acquired daily for 3 days using the high content imaging system InCell 2000 (GE Healthcare), gathering data on neuronal cell size and number, axonal length and neurite branching. Data analysis was performed using the Columbus software (PerkinElmer). Data are presented for 3 days of co-culture. The programme designed for co-culture analysis only takes into account GFP+

cells with at least one projection to exclude counting cell debris. For iMN on iAstrocyte cultures, iAstrocytes were plated in 384-well plates 24 h before plating 1,000 FACS-sorted iMNs. Cultures were maintained for 4 days. Data are presented for 4 days of co-culture.

**Western blot and dot blot analysis.** HEK cells were transfected for 72 h with 650 ng pcDNA6.2-GW/EmGFP-Control-miR-RNAi, pCDNA6.2-GW/EmGFP-human *SRSF1*-miR1 + 2-RNAi or LV-EmGFP-human *SRSF1*-miR1 + 2-RNAi constructs and 50 ng p3XFLAG/human-SRSF1 using 3.5 μg PEI/ml media and one tenth medium volume of OptiMEM. Neuro-2a cells were split into each well of 24-well plates (75,000 cells per well) and transfected for 72 h with 350 ng pcDNA6.2-GW/EmGFP-Control-miR-RNAi (ThermoFisher), pcDNA6.2-GW/EmGFP-mouse SRSF1-miR1 + 2-RNAi, p3xFLAG, p3xFLAG/SRSF1(11-196), p3xFLAG/SRSF1(11-196)-m2 or p3xFLAG/SRSF1(11-196)-m4 and 350 ng pcDNA3.1/RAN-G4C2x38-sense or RAN-C4G2x39-antisense using 3 μg PEI/1 μg DNA and one tenth medium volume OptiMEM.

Proteins were extracted 72 h post-transfection. Cells were washed in ice-cold phosphate-buffered saline (PBS) and scraped into ice-cold lysis buffer (50 mM Hepes pH7.5, 150 mM NaCl, 10% glycerol, 0.5% Triton X-100, 1 mM EDTA, 1 mM DTT, protease inhibitor cocktail (Sigma)). Cells were left to lyse on ice for 10 min followed by centrifugation at 17,000 g at 4 °C for five minutes. Protein extracts were quantified using Bradford Reagent (BioRAD), resolved by SDS–PAGE, electroblotted onto nitrocellulose membrane and probed using the relevant primary antibody. Human/mouse SRSF1/SF2 [1:1,000 dilution] (Cell Signaling #8241), poly-Gly-Pro [1:10,000 dilution] (kindly received from Prof Stuart Pickering Brown) and Histone H3 (Santa Cruz sc-10809) primary antibodies were detected with horseradish peroxidase (HRP)-conjugated rabbit secondary antibody (Promega), while α-tubulin [1:10,000 dilution] (Sigma, clone DM1A), FLAG [1:2,000 dilution] (*Sigma* F1804, clone M2), ALYREF [1:2,000 dilution] (Sigma A9979, clone 11G5) and poly-Gly-Ala [1:500 dilution] (kindly provided from Prof Dieter Edbauer) antibodies were detected using HRP-conjugated mouse secondary antibody (Promega). Uncropped western blot images are shown in Supplementary Figs 11–16. For dot blot analysis, 50 μg total protein extracts prepared in ice-cold lysis buffer were loaded onto a nitrocellulose membrane using a microfiltration apparatus (Biorad), sliced into strips and analysed by immunoblotting.

**Cytoplasmic fractionation.** Neuro-2a cells were split into 6-well plates (2 × 10⁶ cells per well) and transfected for 72 h with 1 μg pcDNA6.2-GW/EmGFP-Control-miR-RNAi, pcDNA6.2-GW/EmGFP-mouse SRSF1-miR1 + 2-RNAi, p3xFLAG/SRSF1(11-196), or p3xFLAG/SRSF1(11-196)-m4 and 1 μg pcDNA3.1/RAN-G4C2x38-sense or RAN-C4G2x39-antisense using 3 μg PEI/1 μg DNA and one tenth medium volume OptiMEM. iNeurons were cultured in 6-well plates and transduced with 5MOI human LV-*SRSF1*-RNAi lentivirus for 5 days.

For the cytoplasmic fractionation, cells were collected in DEPC PBS and pelleted by centrifugation at 800 g for 5 min. Cell pellets were quickly washed with hypotonic lysis buffer (10 mM HEPES pH 7.9, 1.5 mM MgCl₂, 10 mM KCl, 0.5 mM DTT) and lysed in hypotonic lysis buffer containing 0.16 U μl⁻¹ Ribosafe RNase inhibitors (Bioline), 2 mM PMSF (Sigma) and SIGMAFAST Protease Inhibitor Cocktail tablets, EDTA free (Sigma). For flies, *da-GAL4* was used to drive transgene expression in all tissues, and 10 third instar larvae were homogenized and lysed using the same buffer and a dounce homogenizer. All lysates underwent differential centrifugation (1,500 g, 3 min, 4 °C then 3,500 g r.p.m., 8 min, 4 °C and then 17,000 g, 1 min, 4 °C) transferring the supernatants to fresh tubes after each centrifugation. Nuclear pellets obtained after centrifugation at 1,500 g for 3 min were lysed in Reporter lysis buffer (Promega) for 10 min on ice before centrifugation at 17,000 g, 5 min, 4 °C. Total fractions were collected in Reporter lysis buffer containing 0.16 U μl⁻¹ Ribosafe RNase inhibitors (Bioline), 2 mM PMSF (Sigma) and Protease Inhibitors prior to lysis for 10 min on ice before centrifugation at 17,000 g, 5 min, 4 °C. Total and fractionated extracts were added to PureZOL to extract RNA. Equal volumes of total, nuclear and cytoplasmic lysates were subjected to western immunoblotting using SSRP1 and α-tubulin (Neuro-2a), SSRP1 and Actin (iNeurons), or α-tubulin and α-Histone H3 (*Drosophila*) antibodies.

**Quantitative RT–PCR.** For *Drosophila*, total RNA was extracted from crushed larvae or adult flies using 800 μl PureZOL (BioRAD). Lysate was cleared by centrifugation for 10 min at 12,000 g at 4 °C. 200 μl of chloroform was added and tubes were vigorously shaken for 15 s. After 10 min incubation, tubes were centrifuged at 12,000 g for 10 min at 4 °C and supernatants (400 μl) were collected. RNA was precipitated for 30 min at room temperature with 2 μl Glycogen (5 μg μl⁻¹, Ambion) and 500 μl isopropanol and pelleted at 12,000 g for 20 min at 4 °C. Pellets were washed with 70% DEPC Ethanol and re-suspended in 40 μl DEPC water.

For HEK cells, 50,000 cells were split into each well of 24-well plates and transfected with 700 ng pcDNA6.2-GW/EmGFP-Control or human SRSF1-miR-RNAi constructs using 3.5 μg PEI/ml media and one tenth medium volume OptiMEM (ThermoFisher). For iAstrocytes, 20,000 induced neural progenitor cells (iNPCs) were plated in 6-well plates in astrocyte medium. The day after plating, 3 wells were transduced with lentivirus expressing human *SRSF1*-RNAi at an MOI of 5. Total RNA was extracted from HEK cells 72 h after transfection or iAstrocytes 5 days after transduction and RNA extracted using the EZ Total RNA Isolation

Kit (Geneflow). Briefly, cells were washed in DEPC-treated PBS before lysis in the culture dish using the denaturing solution. Lysed cells collected and equal volume extraction buffer added, vigorously shaken, incubated at room temperature for 10 min and then centrifuged for 15 min at 4 °C and 12,000 g. RNA was precipitated using equal volume of isopropanol overnight at − 20 °C, pelleted at 12,000 g, 4 °C for 15 min, washed with 70% DEPC Ethanol and re-suspended in 22.5 μl DEPC water.

All RNA samples were treated with DNaseI (Roche) and quantified using a Nanodrop (NanoDropTechnologies). Following quantification, 2 μg RNA was converted to cDNA using BioScript Reverse Transcriptase (Bioline). qRT–PCR primers were designed using Primer-BLAST[67] (NCBI) and validated using a 1 in 4 serial template dilution series (standard curve with $R^2 > 0.97$). qRT–PCR reactions were performed in duplicate using the Brilliant III Ultra-Fast SYBR Green QPCR Master Mix (Agilent Technologies) on a MX3000P QPCR system (Statagene) using an initial denaturation step, 45 cycles of amplification (95 °C for 30 s; 60 °C for 30 s; 72 °C for 1 min) prior to recording melting curves. qRT–PCR data was analysed using MxPro (Stratagene) and GraphPad Prism (Version 6). The sequences of qPCR primers are provided in Supplementary Note 3 including citations of refs 68–70.

**MTT cell proliferation assay.** Neuro-2a cells were split into 24-well plates (30,000 cells per well). Each plate contained 4 wells with only media to serve as a blank and 4 wells/treatment. Cells were transfected for 72 h with either 500 ng pcDNA3.1, pcDNA3.1/RAN-G4C2x15, RAN-G4C2x38-sense, RAN-C4G2x15 or RAN-C4G2x39-antisense; or 250 ng pcDNA6.2-GW/EmGFP-Control-miR-RNAi, pcDNA6.2-GW/EmGFP-mouse SRSF1-miR1 + 2-RNAi, p3xFLAG/SRSF1(11-196) or p3xFLAG/SRSF1(11-196)-m4 and 250 ng pcDNA3.1, pcDNA3.1/RAN-G4C2x38-sense or RAN-C4G2x39-antisense.

250 mg Thiazolyl Blue Tetrazolum Bromide reagent (MTT) was added to each well and incubated in the dark at 37 °C for 1 h. Cells were subsequently lysed with equal volume MTT lysis buffer (20%SDS, 50% Dimethylformamide (DMF)) and incubated, shaking, at room temperature for 1 h. Absorbance at 595 nm was then assessed with a PHERAstar FS (BMG Labtech). Absorbance data was retrieved using PHERAstar MARS (BMG Labtech).

**Immunofluorescence of rat cortical neurons.** Cortical neurons were isolated, cultured and transfected as described[21]. Briefly, neurons were transfected using Lipofectamine LTX with PLUS reagent according to manufacturer's instructions (Thermofisher; DNA:PLUS:LTX ratio of 1:0.5:0.5 with 2 μg DNA/100,000 cells per cm²). After 6 h, the transfection mix was replaced with conditioned medium. Immunofluorescence staining of rat cortical neurons was performed 72 h after transfection as described[21] with the exception that cells were blocked with 4% goat serum in PBS for 2 h at room temperature and incubated overnight at 4 °C with the V5 antibody [1:1,000 dilution] (ThermoFisher Scientific #R96025) in PBS containing 4% goat serum. Cells were washed three times with PBS containing 4% goat serum and incubated for 1 h with PBS containing 4% goat serum & goat anti-mouse secondary antibody, Alexa Fluor 594 [1:1,000 dilution] (ThermoFisher Scientific). Cells transfected with pcDNA6.2-GW/EmGFP-Control or human SRSF1-miR-RNAi constructs were subsequently stained with Hoechst 33342 for 10 min at room temperature, washed 3 times with PBS and mounted in fluorescence mounting medium (Dako). After incubation in the secondary antibody, cells transfected with p3xFLAG/SRSF1(11-196) or p3xFLAG/SRSF1(11-196)-m4 were washed 3 times with PBS containing 4% goat serum, incubated at room temperature for one hour with PBS containing 4% goat serum & anti-FLAG M2-FITC antibody (10 μg ml⁻¹; Sigma-Aldrich #F4049) and subsequently stained with Hoechst 33342. Cells were then washed 3 times with PBS and mounted.

**Co-Immunoprecipitation.** Cells were split into one 10 cm plates/treatment (1.5 × 10⁶ cells per plate) and transfected with 15 μg p3xFLAG, p3xFLAG/SRSF1(11-196), p3xFLAG/SRSF1(11-196)-m2 or p3xFLAG/SRSF1(11-196)-m4 using 3 μg PEI/1 μg DNA and one tenth medium volume OptiMEM.

Proteins were extracted from Neuro-2a cells 48 h post-transfection. Cells were washed in ice cold PBS, scraped into 500 μl ice cold lysis buffer, passed through a 21G gauge needle 10 times and left to lyse on ice for 10 min. Lysed cells were cleared by centrifugation at 17,000 g at 4 °C for 5 min and protein extracts were quantified using Bradford Reagent. 2 mg total protein in 1 ml lysis buffer was incubated with 30 μl anti-FLAGM2 affinity resin (Sigma A2220) (which had been blocked overnight with 1% BSA in IP lysis buffer) for 2 h at 4 °C on a rotating wheel. Beads were washed 5 times with lysis buffer and eluted in 50 μl IP lysis buffer supplemented with 100 μg ml⁻¹ 3xFLAG peptide (Sigma #F4799) for 30 min at 4 °C on a rotating wheel. 30 μg total protein and 15 μl eluates were subjected to western immunoblotting using FLAG, NXF1 clone 53H8 [1:2,000] (Abcam ab50609) and α-tubulin antibodies.

**RNA:protein UV crosslinking assays.** Recombinant proteins expressed in 1.5 l of *Escherichia coli* BL21 (DE3)-RP (Novagen) cultures were purified on TALON/Cobalt beads (Clontech) in 1 M NaCl containing buffers (Lysis buffer: 50 mM TRIS-HCl pH 8.0, 1 M NaCl, 0.5% Triton X-100; Wash buffer: 50 mM TRIS-HCl pH 8.0, 1 M NaCl, 0.5% Triton X-100, 5 mM imidazole). Elution was achieved with

imidazole (50 mM TRIS-HCl pH 8.0, 500 mM NaCl, 200 mM imidazole) and 50 mM L-Arg/L-Glu mixture to prevent protein precipitation while retaining interaction with RNA and NXF1 (refs 35,36). 2 µg purified proteins were incubated in RNA-binding buffer (15 mM HEPES at pH 7.9, 150 mM NaCl, 0.2 mM EDTA, 5 mM MgCl$_2$, 0.05% Tween 20, 10% glycerol) with 20 nmoles 5′-end $^{32}$P-radiolabelled probes (RNA oligonucleotides purchased from Dharmacon) for 10 min at room temperature and 10 min on ice prior to UV-crosslinking or not (10 min, 1.5 J cm$^{-2}$). Binding reactions were resolved on SDS–PAGE prior to analysis by Coomassie staining and Phosphoimaging.

**RNA immunoprecipitation assays.** Cells were split into one T-175 flask/treatment ($5 \times 10^6$ cells per plate) and transfected with 30 µg p3xFLAG, p3xFLAG/SRSF1(11-196) or p3xFLAG/SRSF1(11-196)-m4 and 10 µg pcDNA3.1/ RAN-G4C2x15-sense, RAN-G4C2x38-sense, RAN-G4C2x15-antisense or RAN-C4G2x39-antisense using 3 µg PEI/1 µg PEI and one tenth volume OptiMEM. 1% formaldehyde was added to the medium of live cells for 10 min 48 h post-transfection. Formaldehyde was quenched with 250 mM Glycine and DEPC-treated PBS-washed cells were scraped into ice cold RNase-free lysis buffer (DEPC-treated water containing 50 mM Hepes pH 7.5, 150 mM NaCl, 10% glycerol, 0.5% Triton X-100, 1 mM EDTA, 1 mM DTT, 1 µl RNase inhibitor, protease inhibitors). Cells were passed through a 21G gauge needle 10 times and left to lyse on ice for 10 min, followed by centrifugation at 17,000 g at 4 °C for five minutes and quantification using Bradford Reagent. 2.5 mg of total protein at a 1 mg ml$^{-1}$ was incubated with 40 µl anti-FLAGM2 affinity resin (which had been blocked overnight with 1% BSA and 5 µl ml$^{-1}$ ssDNA) overnight at 4 °C on a rotating wheel. Beads were washed 5 times with RNase-free lysis buffer. Complexes were reverse cross-linked and eluted from the resin in EZ RNA extraction denaturing buffer (Geneflow) for 1 h at 70 °C, re-suspending the resin every 10 min. The formaldehyde crosslinks were reversed by heating the samples for 1 h at 70 °C and RNA was extracted using PureZOL (for total samples) or the EZ Total RNA Isolation Kit (for eluted complexes) as described in the qRT–PCR section. Extracted RNA samples were re-suspended in 25 µl RNase-free water, DNase-treated and 10 µl input or eluate RNA was converted to cDNA and used for qPCR as described in the qRT–PCR section.

**Visualization of RNA foci and colocalization studies.** To visualize sense RNA foci, RNA fluorescence in situ hybridization (FISH) was performed as described[12] using a 5′ TYE-563-labelled LNA (16-mer fluorescent)-incorporated DNA probe (Exiqon, Inc.). For human post mortem spinal cord tissue, the study was approved by the South Sheffield Research Ethics Committee and informed consent was obtained for all samples. RNA-FISH and immunohistochemistry were performed on formalin fixed paraffin-embedded (FFPE) tissues as in reference[12]. Anti-SRSF1 antibody (Cell Signaling #8241) at a dilution of 1:200. Mounted slides were visualized using a Leica SP5 confocal microscope system and a 63/1.4 oil immersion objective lens. The presence of RNA foci was assessed at high resolution (848 µm$^2$ per image, $393 \times 393$ pixels) using 0.9 µm z-stacks through the entire volume of the cell.

**Statistical analysis of data.** Either one-way or two-way ANOVA (analysis of variance) with Tukey's correction for multiple comparisons was used for most experiments with the following exceptions: DPR analysis in primary neurons used Fisher's exact test; fly climbing ability was analysed by Kruskal-Wallis non-parametric test with Dunn's correction for multiple comparisons; and the analysis of G4C2x36 transcripts in Drosophila used paired two-tailed t-test. No randomization was used in the animal studies. Data were plotted using GraphPad Prism 6. Significance is indicated as follows; NS: non-significant, $P \geq 0.05$; *$P < 0.05$; **$P < 0.01$; ***$P < 0.001$; ****$P < 0.0001$. RNA foci, DPR-positive neurons, crawling and climbing assays were analysed in a blinded manner and several investigators carried out the analysis. Several researchers were involved in producing qRT–PCR and western blot data.

**Data availability.** All data files and files produced for statistical analysis are available on request.

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

## Acknowledgements

We acknowledge Prof Stuart Pickering-Brown and Prof Dieter Edbauer for providing the DPR antibodies. We also thank Prof Stuart A. Wilson and Prof Thomas Jessell for respectively providing the FLAG-tagged SRSF1 plasmids and the Hb9:GFP mouse stem cells. This work is supported by the Motor Neurone Disease Association grant Hautbergue/Apr16/846-791 (G.M.H., L.F., A.J.W., P.J.S., L.M.C.) and the Medical Research Council (MRC) grant MR/M010864/1 (K.N., G.M.H., P.J.S.) in collaboration with Professor Jun Xu (Projects of International Cooperation and Exchanges NSFC: 81261130318) at Tongji University in China. L.F. and G.M.H. are also supported by the Thierry Latran Foundation (grant FTLAAP2016/FERRAIUOLO/Astrocyte secretome). G.M.H. and L.F. were also respectively supported by The Royal Society (grant RG140690) and a Marie Curie Fellowship (grant 303101). Y.H.L. is supported by a Postdoctoral Research Abroad Program sponsored by the Taiwanese Ministry of Science and Technology (105-2917-I-564-070). J.E.D. was supported by the University of Sheffield Moody Family Endowment (CPW) and the Motor Neurone Disease Association (grant Hautbergue/Mar16/900-790). E.K. was supported by Eve Davis Studentship. E.F.S. is supported by a Motor Neurone Disease Association Prize Studentship (grant DeVos/Oct13/870-892). K.J.D.V. is supported by the Thierry Latran Foundation (Project RoCIP) and Medical Research Council (MRC) grants MR/K005146/1 and MR/M013251/1. M.A. is supported by a European Research Council (ERC) grant (GTNCTV–294745). A.J.W. is supported by MRC core funding (MC-A070-5PSB0) and ERC Starting grant (309742). This work was also supported in part by the European Community's Seventh Framework Programme [FP7/2007-2013] under the EuroMOTOR project [grant agreement no 259867 to J.K. and P.J.S.]; by a Motor Neuron Disease Association/Medical Research Council Lady Edith Wolfson Fellowship award [MR/K003771/1 to J.C.-K.]; and by the Sheffield Hospitals Charitable Trust [grant 131425 to M.J.S.]. P.J.S. is supported as a National Institute for Health Research Senior Investigator.

## Author contributions

G.M.H. conceived the study. G.M.H. and A.J.W. designed and supervised experiments. G.M.H., L.M.C., L.F., J.C.K., A.H., Y.H.L., C.S.B., J.E.D., M.A.M., P.G., M.J.S., E.F.S. and K.J.D.V. performed *in vitro* experiments. A.S.M., S.M.A. and A.J.W. performed *Drosophila* assays. J.S.C., E.K. and M.A. produced lentivirus. P.J.S. oversaw and supervised the study. G.M.H, A.J.W. and P.J.S. wrote the manuscript with input from all authors.

## Additional information

**Competing interests:** G.M.H., M.A., A.J.W. and P.J.S. have filed a patent application on the use of SRSF1 antagonists for the treatment of neurodegenerative disorders. The remaining authors declare no competing financial interests.

