## [Peer Review File · Nature Communications]

Reviewers' comments:

Reviewer #1 (Remarks to the Author):

There are three proposed pathomechanisms associated with G4C2 repeat expansions in ALS/FTD: (i) RNA toxic gain-of-function by sequestration of RNA-binding factors, (ii) protein toxic gain-of-function due to repeat-associated non-ATG (RAN) translation that occurs in sense and antisense reading frames to produce five dipeptide-repeat proteins (DPRs) and (iii) haploinsufficiency due to decreased expression of the C9ORF72 protein. Here the authors have built on their prior findings published in *Brain* 137, 2040-2051 (2014) where they identified several proteins interacting with (G4C2)₅ repeats, including SRSF1 (serine/arginine-rich splicing factor 1) and ALYREF (Aly/REF export factor). They hypothesize that binding of SRSF1/ALYREF to G4C2 repeats increases binding of NXF1, which overrides normal nuclear retention mechanisms causing nuclear export of repeat containing transcripts and translation of DPRs. To support this they have shown that downregulation of SRSF1 prevents neurodegeneration and locomotor deficits in *Drosophila* (G4C2)₃₆, and death of mouse motor neurons co-cultured with C9-astrocytes in which SRSF1 has been downregulated. They also show that downregulation of SRSF1 or expression of a mutant SRSF1 that does not interact with NXF1, impede export of G4C2 containing transcripts leading to reduced DPR expression.

This is an interesting study, however there are a number of concerns.

Major comments

1) The authors are working on the hypothesis that:

'Our structural and functional analysis of the interactions with RNA and the NXF1 (nuclear export factor 1) receptor, which mediates mRNA nuclear export, previously revealed that export adaptors remodel NXF1 to increase its affinity for mature mRNAs, preventing in turn the nuclear export of unprocessed transcripts¹⁴⁻¹⁷. Here, we hypothesized that: (i) Excessive binding of nuclear export adaptor(s) onto G4C2-repeat transcripts will force interactions with NXF1 and override the normal nuclear retention mechanisms'

This argument requires more explanation. What are the normal nuclear retention mechanisms? How does this apply to G4C2 repeat transcripts? The implication is that the repeat transcripts are over-riding the nuclear retention machinery allowing export of unprocessed transcripts. However, fully mature repeat containing mRNAs (not unprocessed transcripts) have been reported (Niblock 2016), and increased nuclear retention of repeat transcripts (which is proposed here as a therapeutic strategy) would increase RNA foci, and in turn cause toxicity through sequestration of RNA binding proteins. Indeed, the authors show that downregulation of SRSF1 leads to increased nuclear RNA foci in astrocytes, and although this led to reduction in neurotoxic effects in co-cultures of these astrocytes with mouse motor neurons, the effect of downregulation of SRSF1 in C9-iPSC-derived motor neurons (a more relevant model) was not shown.

2) From the original publication by the authors in *Brain*, SRSF1 was shown to bind to (G4C2)₅ repeats in biotinylated RNA pulldown assays but did not co-label RNA foci in the cerebellar granule layer or the ventral horn. This suggests that SRSF1 does not bind G4C2 repeats in disease relevant tissues. Does SRSF1 label RNA foci in the current study?

3) The rationale for studying effects of SRSF1 downregulation in astrocytes is not at all clear. RNA foci are a pathological feature in neurons of C9 cases and the glial phenotype alluded to by the authors in Ref 23 is in microglia of C9 KO mice (O'Rourke et al 2016). It would be more relevant to study the effect of downregulating SRSF1 using C9-iPSC-derived motor neurons.

Minor

4) Comment is made that SRSF1 is non-essential (e.g. abstract), this seems very unlikely and down regulation of SRSF1 could lead to changes in RNA metabolism.

5) Page 3, 2nd Para, First sentence 'Intron-containing G4C2-repeat transcripts escape the protective mechanism of nuclear retention to become translated into DPRs in the cytoplasm.'

References reporting G4C2-repeat mRNAs should be included.

What does it mean 'escape the protective mechanism of nuclear retention'? What is the evidence for this? Retained G4C2 transcripts would presumably lead to more nuclear RNA foci, and as such increased toxicity through sequestration of RNA-binding factors.

6) Page 4, 1st Para

'Consistent with the previous results, SRSF1-depletion restored locomotor function while ALYREF-RNAi provided no rescue of the behavioural phenotype (Fig. 1c-d).'

This was not a rescue experiment but prevention through crossing with RNAi flies

7) Abstract: It is not appropriate to call the astrocyte co-cultures a 'state-of-the art-assay'

8) Following comment in the abstract seems unnecessary, too generalized and overstated: 'We anticipate these discoveries to form a robust foundation for additional studies across the fundamental and translational fields of research for the identification of altered biological pathways causing neurodegeneration and the development of gene therapy and drug screening programs.'

8) Figure 3B: the authors measure cytoplasmic/total GGGGCC mRNA to demonstrate lower cytoplasmic levels in SRSF1 depletion and SRSF1-m4 conditions. Are total levels (not just ratio) similar between conditions?

Reviewer #2 (Remarks to the Author):

Among the pathogenic mechanisms to explain C9ORF72-linked ALS/FTD (frontotemporal dementia), two prominent hypotheses are the gain of toxicities from the repeat-associated non-ATG production of toxic dipeptide-repeat proteins and the RNA toxicities from abnormally expanded repeats of hexanucleotides, GGGGCC (G4C2). In this study, authors demonstrated that partial depletion of SRSF1 protein ameliorated neurodegeneration in a *Drosophila* model of C9ORF72-related ALS/FTD. Depletion of SRSF1 also showed neuroprotection in co-culture system with motor neurons and ALS patient-derived astrocytes. Moreover, they showed that depleting SRSF1 or inhibiting its interaction with NXF1 impedes the nuclear export of C9ORF72 repeat RNAs and the toxic dipeptide-repeat proteins from G4C2 repeats. Although the recent studies reported the nuclear export/import was affected in C9ORF72-linked ALS/FTD, their study identified the molecular pathway to manipulate the RNA nuclear export pathway for ameliorating C9ORF72-related ALS/FTD for the first time. Overall, this study is well designed and executed. It holds a novelty and makes an impact on the ALS/FTD research fields. The referee has some concerns and comments as below.

1. Authors identified SRSF1, the non-essential nuclear export protein as a binding partner for C9 repeat RNA. It is of interest to know how SRSF1-binding to C9 RNAs prevents normal splicing of intronic pre-mRNAs (including abnormal G4C2 repeats). If the splicing machinery is intact, G4C2

repeats are spliced out during mature mRNA production in the nucleus, then nuclear export of abnormal repeat containing RNAs may cause a minimum impact.

2. Figure 2: Using the co-culture experiment using mouse wild-type motor neurons (with HB9-GFP expression) and induced astrocytes derived from C9ORF-ALS patients, a partial depletion of SRSF1 in the patient-derived astrocytes promoted the survival of mouse motor neurons and reduced cytoplasmic RNA foci in induced astrocytes. Do the authors think preventing nuclear exports of C9-mRNAs in the astrocytes is a key component of pathomechanism? In other words, non-cell autonomous toxicities from C9-astrocytes rather than cell autonomous mechanism in the neurons are more crucial?

3. Related to the previous comment, the effect of SRSF1 knockdown on RNA foci or dipeptide repeats in the C9-expressing motor neurons was not evaluated. According to the results from C9-fly model, SRSF1 is participated in the multiple cell types such as neurons and glial cells. The role of the SRSF1 in the motor neurons should be evaluated.

4. The authors' results suggest that cytoplasmic accumulation of C9 transcripts and dipeptides seems to be a key component of neurotoxicity. The referee feels it is worth discussing the results from other C9-ALS/FTD studies regarding this issue (e.x. Freibaum et al., Nature, 2015; Zhang et al., Nature, 2015, Jovicic et al., Nat Neurosci, 2015, Boeyanems et al., Sci Rep, 2016, Zhang et al., Nat Neurosci, 2016). On the other hand, in the other repeat-related diseases such as polyglutamine diseases or spinocerebellar ataxia, accumulation of the mutant products in nucleus seems to be toxic. Authors are encouraged to discuss the points.

5. Ethical statement is missing. Do the authors have an approval from institutional review board to perform the research using the patient-derived cells?

Reviewer #3 (Remarks to the Author):

The manuscript by Hautbergue et al., proposes that interfering with the SRSF1-NXF1 nuclear export pathway prevents toxicity of the G4C2 and C4G2 repeats in flies and in cell culture by preventing nuclear export of the expansion RNAs. This hypothesis is largely based on previous data showing SRSF1 binds to G4C2 expansion transcripts in pull-down assays and in co-localization studies. Overall the authors show convincing evidence that RNAi knockdown of SRSF1 prevents neurodegeneration of fly eyes expressing 36 G4C2 repeats. Additionally, the authors show data that SRSF1 depletion suppresses C9orf72 astrocyte mediated toxicity - and that this decreased toxicity is associated with increased levels of nuclear RNA foci. In Figure 3 the authors extend these results using an overexpression strategy to express G4C2 and C4G2 antisense expansion RNAs. Again, cytoplasmic localization of the RNAs decreases SRSF1 depletion and similar effects are seen when SRSF1 mutations that block the interaction with NXF1 are expressed. While overall this is a strong manuscript, additional data are needed to support the central hypothesis, which is that SRSF1 binds to the C9 expansion RNAs and facilitates their cytoplasmic export. My specific comments follow:

1. Previous studies only provide evidence that the G4C2 expansion RNAs, not the C4G2 antisense RNAs bind to and co-localize with SRSF1. The antisense CCCC GG RNA has a very different motifs and structure compared to the GGGGCC repeat RNAs, which therefore may not bind to or sequester the same RNA binding protein. This raises questions about the model. The authors should test if CCCC GG binds to SRSF1 and also if it colocalizes with the antisense RNA foci. The best strategy to do this would be CLIP.

2. If binding of C4G2 expansion RNAs to SRSF1 does not occur then the central model does not fit well with the data as currently interpreted. If C4G2 RNAs do not bind SRSF1 it will be important to understand why these two different plasmids show similar results.

3. Since GP expression would occur from both sense and antisense directions, the authors should test if the reduction in toxicity and GP expression is caused by nuclear retention of antisense (G4C2 expansion RNAs expressed from C4G2 constructs). Strand specific RT-PCR or epitope tagged proteins could help determine this.

4. As mentioned, SRSF 1 is also a splicing factor. So it is important to show the C9orf72 intron 1 splicing pattern after applying SRSF1 RNAi in the iPSC cells because the retention of the GGGGCC repeat containing intron will largely affect the nuclear-cytoplasmic localization of these repeats. Although SRSF 1 RNAi in cells transfected with GGGGCC constructs without splicing site reduced RAN translation, this does not exclude the possibility that both nuclear export and RNA splicing contribute to the cytoplasmic localization and RAN translation in patient derived cells. This should be addressed in the manuscript.

5. The authors tried to connect the NXF1 export receptor to the SRSF1 pathway by using the SRSF1-m4 point mutation, which does not interact with NXF1. Overexpression of this mutation leads to decreased cytoplasmic expansion RNAs and decreased RAN translation, but this could also be due to the mutation affecting nuclear export or splicing activity of SRSF1. It is important to determine how the cytoplasmic expansion RNA and RAN protein levels are affected when knocking down NXF1 to fully clarify the pathway.

Figure 2 c is not clear as described and needs further explanation better images and a better description.

Reviewer #4 (Remarks to the Author):

Hautbergue et al demonstrate that knockdown of SRSF1 suppresses toxicity of hexanucleotide repeat expression (G4C2 and C4G2) and suggest that this may be a therapeutic strategy for C9ORF72-related ALS. Using a Drosophila model, patient astrocyte/neuron cocultures, and transfected N2A cells, they conclude that SRSF1 is required for nuclear export of G4C2 and C2G4 RNA repeats. Finally, they use a dominant-negative SRSF1 construct to show that nuclear export of C9ORF72 repeat transcripts and subsequent RAN translation depends on the interaction of SRSF1 with the NXF1 export receptor. If the data were convincing, this would be an extremely important finding. Unfortunately, though, the authors show limited data from multiple different model systems without many necessary controls, leaving the reader with an incomplete understanding of the effects of SRSF1 depletion in each system. The authors draw broad conclusions about mechanism from pieces of data obtained from very different model systems, and do not consider alternative explanations. SRSF1 regulates multiple aspects of RNA metabolism including transcription, stability, NMD, and translation. Importantly, an alternative explanation for suppression of their phenotypes in most experiments is a reduction in transcription or translation of the G4C2 repeats—while the authors quantitate RNA foci, they do not quantitate total message using RT-PCR, and RAN translation is only tested in an artificial system using transfected N2A cells.

In order to convince the reader that SRSF1 knockdown is truly a potential therapeutic strategy, the authors must test their hypothesis in a mammalian system in which the repeats are expressed in the context of the endogenous c9orf72 gene. In this regard, experiments shown in figure 2d-e (using iAstrocytes and motor neuron co-cultures) come closest to testing their hypothesis, but the data provided are quite limited and unconvincing. First, how does SRSF1 knockdown in astrocytes prevent nonautonomous toxicity to motor neurons? This should be the ideal system to test whether SRSF1 knockdown inhibits DPR production (and presumably secretion into the media?),

yet DPR production is not tested in this system.

Second, the authors state that they use this coculture assay to look at nonautonomous astrocyte-mediated toxicity because iNeurons from c9ALS patients do not have decreased survival. However, multiple groups have now shown that iPS-derived neurons from c9-ALS patients have abnormal RNA profiles that mimic those seen in human tissue in addition to multiple cellular phenotypes (e.g. nuclear transport abnormalities) and increased sensitivity to stress. The authors should determine the effects of SRSF1 knockdown on neurons derived from c9orf72 patients and controls. An alternative approach if the authors cannot detect neuronal phenotypes in patient-derived iNeurons would be to look at the effect of SRSF1 knockdown on phenotypes recently described in BAC transgenic mice.

Another major question/concern is whether SRSF1 knockdown specifically inhibits G4C2 hexanucleotide RNA export or whether it has a global effect on mRNA export/transcription. The authors measure G4C2 mRNA export relative to U1 snRNA in Fig 3b in transfected N2A cells, but it would be important to perform similar experiments in iNeurons/astrocytes where they can determine the effect of SRSF1 knockdown on transcription/splicing/export of individual c9orf72 isoforms in addition to control genes. Does SRSF1 inhibit export of the entire c9orf72 mRNA with a retained G4C2-containing intron, or does it prevent export of spliced introns? At a minimum, the authors should measure total mRNA in nucleus vs cytoplasm as was performed in Friebaum et al, Nature 2015 (where the authors showed a general reduction in polyA+ export in C9 iPSNs).

Moreover, the analysis of the effects of SRSF1 knockdown in *Drosophila* is very superficial. There are many simple yet critical experiments that were omitted.

1) A critical control experiment is to determine if SRSF1 knockdown suppresses toxicity caused by overexpression of DPRs ("protein only" GR or PR). The authors claim to show in extended data fig. 7 that SRSF1-RNAi doesn't significantly suppress the climbing phenotype caused by Arginine-containing DPRs even though there does appear to be some rescue (but doesn't reach statistical significance—surprisingly, they only test 8 flies for GR36!). I suspect with increasing N (to level of control), they will see significant rescue. An easier experiment, though, is just to look for suppression of the rough eye phenotype caused by GR36 and PR36. If SRSF1-RNAi suppresses the GR36 and PR36 rough eye phenotype, this would suggest a general inhibition of transcription/export rather than being specific to G4C2 repeats.

2) The fly model used has serious caveats when it comes to analysis of G4C2 RNA export since it contains a polyA tail after the repeats. To show that SRSF1 knockdown specifically inhibits G4C2 export (rather than globally reducing transcription or export), the authors should assess G4C2 levels (total, nuclear, and cytoplasmic) in addition to that of control mRNAs. They should also test the effects in the "intron model" developed by the Gao lab in which they show correlation of toxicity with nuclear export and RAN translation.

Finally, the authors propose that the mechanism by which the G4C2 repeats export the nucleus is via a direct interaction with SRSF1. This model would lead to several predictions. First, increasing SRSF1 should enhance export and therefore toxicity. Second, mutating SRSF1 to specifically prevent interaction with G4C2 repeats should also suppress export and toxicity. These experiments should be performed to test their proposed mechanism.

Other considerations:

The submission appears to be a Nature letter format with only 3 figures rather than a Nature Communications article format.

Extended data Figure 1 and 5 are almost identical – (human vs mouse). One extended figure to show generation of this construct is more than sufficient.

Minor points:

Figure 2c legend – authors should state what green fluorescence is (Hb9-GFP+ MNs).

Point-by-point response to reviewers' comments

We acknowledge the reviewers for positive and constructive comments. We have addressed the vast majority of the questions and concerns having performed additional experiments to confirm and validate that antagonizing the SRSF1 function specifically inhibits the nuclear export of pathological *C9ORF72* repeat transcripts but not the splicing of the hexanucleotide repeat expansions. We now also show that the SRSF1 depletion does not affect the expression levels, splicing or nuclear export of wild-type spliced transcripts required for the production of the *C9ORF72* protein. Moreover this intervention does not affect the morphology of motor neurons derived from *c9ORF72*-ALS patients. Some experiments have been performed in new models including primary neurons and neurons-derived from control and *C9ORF72*-ALS patients as requested by reviewers. The revised manuscript now contains 8 figures of data (compared to 3 in the original submission), a schematic model, and 10 Supplementary figures (compared to 8 in the original submission).

Reviewers' comments:

Reviewer #1 (Remarks to the Author):

There are three proposed pathomechanisms associated with G4C2 repeat expansions in ALS/FTD: (i) RNA toxic gain-of-function by sequestration of RNA-binding factors, (ii) protein toxic gain-of-function due to repeat-associated non-ATG (RAN) translation that occurs in sense and antisense reading frames to produce five dipeptide-repeat proteins (DPRs) and (iii) haploinsufficiency due to decreased expression of the *C9ORF72* protein. Here the authors have built on their prior findings published in *Brain* 137, 2040-2051 (2014) where they identified several proteins interacting with (G4C2)₅ repeats, including SRSF1 (serine/arginine-rich splicing factor 1) and ALYREF (Aly/REF export factor). They hypothesize that binding of SRSF1/ALYREF to G4C2 repeats increases binding of NXF1, which overrides normal nuclear retention mechanisms causing nuclear export of repeat containing transcripts and translation of DPRs. To support this they have shown that downregulation of SRSF1 prevents neurodegeneration and locomotor deficits in *Drosophila* (G4C2)₃₆, and death of mouse motor neurons co-cultured with *C9*-astrocytes in which SRSF1 has been downregulated. They also show that downregulation of SRSF1 or expression of a mutant SRSF1 that does not interact with NXF1, impede export of G4C2 containing transcripts leading to reduced DPR expression.

This is an interesting study, however there are a number of concerns.

Major comments

1) The authors are working on the hypothesis that:

'Our structural and functional analysis of the interactions with RNA and the NXF1 (nuclear export factor 1) receptor, which mediates mRNA nuclear export, previously revealed that export adaptors remodel NXF1 to increase its affinity for mature mRNAs, preventing in turn the nuclear export of unprocessed transcripts¹⁴⁻¹⁷. Here, we hypothesized that: (i) Excessive binding of nuclear export adaptor(s) onto G4C2-repeat transcripts will force interactions with NXF1 and override the normal nuclear retention mechanisms'

This argument requires more explanation. What are the normal nuclear retention mechanisms? How does this apply to G4C2 repeat transcripts? The implication is that the repeat transcripts are over-riding the nuclear retention machinery allowing export of unprocessed transcripts.

The manuscript has now been rewritten in the *Nature Communications* format allowing for a comprehensive introduction. Pre-mRNA transcripts are retained in the nucleus until

co-transcriptional capping, splicing and poly-adenylation has occurred. The nuclear export of mature transcripts has been coupled to co-transcriptional processing via the TREX (Transcription-Export complex) (Strässer et al., 2002) which is recruited onto pre-mRNA during splicing in vertebrates (Masuda et al., 2005). We previously showed that the TREX complex is a binding platform for the NXF1 mRNA nuclear export receptor (Viphakone et al., 2012) and that the nuclear export of mRNA involves the structural remodeling of NXF1 by nuclear export adaptor proteins, such as ALYREF and SRSF1 (Hautbergue et al., 2008), in concert with a subunit of the TREX complex (Viphakone et al., 2012). In the absence of interactions with a nuclear export adaptor and the TREX complex, NXF1 adopts a closed conformation which silences its RNA-binding activity. However, these interactions trigger the opening of NXF1 and expose its RNA-binding domain. We proposed that the RNA-binding affinity regulation of NXF1 during co-transcriptional processing offers a control mechanism to retain pre-mRNA in the nucleus (Hautbergue et al., 2008; Walsh et al., 2010). We have now detailed and referenced these mechanisms in the introduction.

In relation to G4C2 repeat transcripts, we hypothesized that the intronic repeat hexanucleotide expansions which sequesters RNA-binding proteins including nuclear export adaptors ALYREF and SRSF1 (Cooper-Knock et al., 2014; Lee et al., 2013) might trigger the high RNA-affinity remodeling of NXF1 due to increased local concentrations and the abnormal nuclear export of intron-1-containing repeat transcripts that were recently characterised in a small proportion in the cytoplasmic compartment (Niblock et al., 2016).

However, fully mature repeat containing mRNAs (not unprocessed transcripts) have been reported (Niblock 2016), and increased nuclear retention of repeat transcripts (which is proposed here as a therapeutic strategy) would increase RNA foci, and in turn cause toxicity through sequestration of RNA binding proteins.

The Niblock 2016 paper was not in press at the time of submission but has now been cited. Indeed, this study also shows that a small proportion of *C9ORF72* transcripts retaining repeat expansions in intron-1 are found in the cytoplasm (Niblock et al., 2016). However, Tran et al reported in *Neuron* (Tran et al., 2015) a thorough and compelling study that showed a 10-fold increase in the nuclear RNA foci does not induce neurotoxicity effects. This is in complete agreement with our study in which nuclear accumulation of *C9ORF72* transcripts retaining pathological repeat expansions in intron-1 confers neuroprotection in *SRSF1*-RNAi-treated neurons-derived from *C9ORF72*-ALS patients (Figure 8).

Indeed, the authors show that downregulation of SRSF1 leads to increased nuclear RNA foci in astrocytes, and although this led to reduction in neurotoxic effects in co-cultures of these astrocytes with mouse motor neurons, the effect of downregulation of SRSF1 in C9-iPSC-derived motor neurons (a more relevant model) was not shown.

We appreciate this suggestion and we now show a neuroprotective effect of the SRSF1 depletion in motor neurons derived from *C9ORF72*-ALS patients (Fig. 8c).

2) From the original publication by the authors in *Brain*, SRSF1 was shown to bind to (G4C2)₅ repeats in biotinylated RNA pulldown assays but did not co-label RNA foci in the cerebellar granule layer or the ventral horn. This suggests that SRSF1 does not bind G4C2 repeats in disease relevant tissues. Does SRSF1 label RNA foci in the current study?

We have tried to investigate the potential co-localization of SRSF1 with sense (in the *Brain* study, (Cooper-Knock et al., 2014)) or antisense RNA foci (in another study, (Cooper-Knock et al., 2015)) in motor neurons from *C9ORF72*-ALS post-mortem tissues. We have indeed detected some nuclear co-localisation events in motor neurons from spinal cord and motor cortex. However, these might have been co-incidental due to the diffuse nuclear staining of SRSF1. We have therefore preferred not to publish these data which remain in our view inconclusive.

On the other hand, we consider that the potential co-localisation of SRSF1 with RNA foci is irrelevant in the context of nuclear export since RNA foci are unlikely to be exported through the nuclear pore due to their size. Single mRNA molecules require unfolding/unwinding by RNA helicases on both the nuclear and cytoplasmic sides for their passage through the nuclear pore. Moreover, while co-localisation may be a useful indicator of a functional interaction, we have demonstrated both their physical interaction and a functional outcome by disrupting this interaction.

3) The rationale for studying effects of SRSF1 downregulation in astrocytes is not at all clear. RNA foci are a pathological feature in neurons of C9 cases and the glial phenotype alluded to by the authors in Ref 23 is in microglia of C9 KO mice (O'Rourke et al 2016). It would be more relevant to study the effect of downregulating SRSF1 using C9-iPSC-derived motor neurons.

Astrocyte-mediated toxicity in ALS is a key mechanism driving motor neuron death and astrocytes carrying *C9ORF72* mutations are not an exception (Meyer et al., 2014). It has also been demonstrated that poly-dipeptides encoded by the *C9ORF72* repeat transcripts impede RNA biogenesis and alter the splicing of key transcripts. e.g. EAAT2, in astrocytes, thus altering cell function (Kwon et al., 2014). There are now multiple lines of evidence indicating that ALS pathology involves both neurons and glia and most likely effective therapies will have to be tested in and target both cell types.

However, we acknowledge that it is also important to investigate the potential effect of SRSF1 depletion in motor neurons derived from C9ORF72-ALS patients. As reported in (Donnelly et al., 2013; Sareen et al., 2013) and Fig. 8a,b in this manuscript, neurons derived from C9ORF72-ALS patients do not exhibit any morphological abnormalities or impaired growth compare to neurons-derived from control patients. However, we have extended our study to address this and now show that the depletion of SRSF1 in motor neurons derived from C9ORF72-ALS patients significantly improves their survival in co-cultures with astrocytes derived from C9ORF72-ALS patients (Fig. 8c) demonstrating a neuroprotective role of the SRSF1 depletion in the disease relevant cells.

Minor

4) Comment is made that SRSF1 is non-essential (e.g. abstract), this seems very unlikely and down regulation of SRSF1 could lead to changes in RNA metabolism. We have removed "non-essential". However, it is important to point out that a number of studies, including (Hautbergue et al., 2009; Katahira et al., 2009), highlighted that the depletion of a single nuclear export adaptor is not sufficient to alter the growth or nuclear export of global mRNAs in human cells. The individual knockdown of each of the seven SRSF1-7 proteins recently demonstrated that the nuclear export of only a small proportion of transcripts (<0.5-2% mRNAs) is affected, clearly highlighting redundancy and/or cooperation in the NXF1-dependent nuclear export adaptor function (Müller-McNicoll et al., 2016).

5) Page 3, 2nd Para, First sentence 'Intron-containing G4C2-repeat transcripts escape the protective mechanism of nuclear retention to become translated into DPRs in the cytoplasm.' References reporting G4C2-repeat mRNAs should be included. As aforementioned in the major comment 1 response, we have now included a large introduction providing mechanistic details on the nuclear retention of intron-containing transcripts. Here we also show that the sequestration of SRSF1 on hexanucleotide repeats triggers the nuclear export of repeat transcripts independently of functional coupling to pre-mRNA splicing (Fig. 6). The reference Niblock et al. 2016 has also been cited.

What does it mean 'escape the protective mechanism of nuclear retention'? What is the evidence for this? Retained G4C2 transcripts would presumably lead to more nuclear RNA foci, and as such increased toxicity through sequestration of RNA-binding factors. Again, as mentioned in our answer to major comment 1, a 10-fold increase in the nuclear RNA foci does not induce neurotoxicity effects (Tran et al., 2015) in full agreement with

this study. The contribution of RNA-mediated toxicity by sequestration of RNA-binding factors and of DPR-mediated neurotoxicity remains to be fully evaluated in C9ORF72-ALS. This study supports DPR-mediated neurotoxicity is predominant over the accumulation of nuclear RNA foci/ retention of C9ORF72 repeat transcripts consistent with other reports ((Mizielinska et al., 2014; Tran et al., 2015) for examples).

6) Page 4, 1st Para

'Consistent with the previous results, SRSF1-depletion restored locomotor function while ALYREF-RNAi provided no rescue of the behavioural phenotype (Fig. 1c-d).'

This was not a rescue experiment but prevention through crossing with RNAi flies.

“while ALYREF-RNAi provided no rescue” was replaced by “while ALYREF-RNAi showed no effect”.

7) Abstract: It is not appropriate to call the astrocyte co-cultures a 'state-of-the-art assay'.

We have removed state-of-the-art assay from the text.

8) Following comment in the abstract seems unnecessary, too generalized and overstated: 'We anticipate these discoveries to form a robust foundation for additional studies across the fundamental and translational fields of research for the identification of altered biological pathways causing neurodegeneration and the development of gene therapy and drug screening programs.'

We have removed this sentence from the manuscript.

8) Figure 3B: the authors measure cytoplasmic/total GGGGCC mRNA to demonstrate lower cytoplasmic levels in SRSF1 depletion and SRSF1-m4 conditions. Are total levels (not just ratio) similar between conditions?

This is a fair point and we thank the reviewer for highlighting this. Total levels were indeed quantified (as used for calculating ratios) and were not affected ruling out major effects of SRSF1 depletion on the transcription or the stability of C9ORF72 repeat transcripts. We now show the total and cytoplasmic levels of C9ORF72 repeat transcripts in Supplementary Fig. 7a and the cytoplasmic/total ratio in Fig. 6f. In addition, we also obtained similar data in G4C2x36+SRSF1-RNAi *Drosophila* (total and cytoplasmic levels shown in Supplementary Fig. 7b and cytoplasmic/total ratio shown in Fig. 6h) and in SRSF1-RNAi treated neurons-derived from C9ORF72-ALS patients (Fig. 8g,h).

Reviewer #2 (Remarks to the Author):

Among the pathogenic mechanisms to explain C9ORF72-linked ALS/FTD (frontotemporal dementia), two prominent hypotheses are the gain of toxicities from the repeat-associated non-ATG production of toxic dipeptide-repeat proteins and the RNA toxicities from abnormally expanded repeats of hexanucleotides, GGGGCC (G4C2). In this study, authors demonstrated that partial depletion of SRSF1 protein ameliorated neurodegeneration in a *Drosophila* model of C9ORF72-related ALS/FTD. Depletion of SRSF1 also showed neuroprotection in co-culture system with motor neurons and ALS patient-derived astrocytes. Moreover, they showed that depleting SRSF1 or inhibiting its interaction with NXF1 impedes the nuclear export of C9ORF72 repeat RNAs and the toxic dipeptide-repeat proteins from G4C2 repeats. Although the recent studies reported the nuclear export/import was affected in C9ORF72-linked ALS/FTD, their study identified the molecular pathway to manipulate the RNA nuclear export pathway for ameliorating C9ORF72-related ALS/FTD for the first time. Overall, this study is well designed and executed. It holds a novelty and makes an impact on the ALS/FTD research fields. The referee has some concerns and comments as below.

1. Authors identified SRSF1, the non-essential nuclear export protein as a binding partner for C9 repeat RNA. It is of interest to know how SRSF1-binding to C9 RNAs prevents normal splicing of intronic pre-mRNAs (including abnormal G4C2 repeats). If the splicing machinery is intact, G4C2 repeats are spliced out during mature mRNA

production in the nucleus, then nuclear export of abnormal repeat containing RNAs may cause a minimum impact.

The hexanucleotide repeat expansions are located in intron-1 of the *C9ORF72* gene. It was reported that the proportion of intron-1 spliced transcripts was not affected in *C9ORF72*-ALS post mortem brain tissue and neurons-derived from *C9ORF72*-ALS patients (Tran et al., 2015). We have obtained similar results in this study showing that the proportion of intron-1-spliced *C9ORF72* transcripts is not affected in neurons-derived from *C9ORF72*-ALS patients compared to control patients (Fig. 8e,f). Moreover the depletion of SRSF1 does not affect the splicing of intron-1 (Fig. 8e,f). On the other hand, while intron-1-retaining *C9ORF72* transcripts were not detected in the neuronal cytoplasm of control patients in agreement with normal nuclear retention mechanisms, the presence of pathological repeat expansions in intron-1 of the *C9ORF72* gene specifically licenses repeat transcripts for nuclear export (Fig. 8g,h). We further showed that the depletion of SRSF1 specifically inhibits the nuclear export but not the expression levels (Fig. 8g,h) or intron-1-splicing of *C9ORF72* repeat transcripts (Fig. 8e,f).

2. Figure 2: Using the co-culture experiment using mouse wild-type motor neurons (with HB9-GFP expression) and induced astrocytes derived from *C9ORF72*-ALS patients, a partial depletion of SRSF1 in the patient-derived astrocytes promoted the survival of mouse motor neurons and reduced cytoplasmic RNA foci in induced astrocytes. Do the authors think preventing nuclear exports of C9-mRNAs in the astrocytes is a key component of pathomechanism? In other words, non-cell autonomous toxicities from C9-astrocytes rather than cell autonomous mechanism in the neurons are more crucial? We think that astrocyte-mediated toxicity is involved in ALS pathogenesis but the exact contribution of non-cell autonomous toxicities is not clearly defined. We are not sure at this point how the inhibition of the nuclear export of *C9ORF72* repeat transcripts in astrocytes confers motor neuron neuroprotection. Exosomes are known to contain RNA molecules. It is possible that the SRSF1 depletion affects the nuclear export and/or the expression of RNA involved in regulating neuronal homeostasis/death pathways. It might also be that DPRs produced in astrocytes are packaged into the exosomal extra-cellular vesicles and exacerbate neurotoxicity when delivered to the motor neurons. Consistent with this, it was also reported that the expression of DPRs in astrocytes alter their gene expression profile thereby inducing a change in the content of the exosomes (Kwon et al., 2014) that would be delivered to motor neurons.

However, we have now performed the reciprocal experiment showing that the depletion of SRSF1 in motor neurons derived from *C9ORF72*-ALS patients significantly improves their survival in co-cultures with astrocytes derived from *C9ORF72*-ALS patients (Fig. 8c) demonstrating a neuroprotective role of the SRSF1 depletion both in non-cell autonomous toxicity and in the disease relevant cells.

3. Related to the previous comment, the effect of SRSF1 knockdown on RNA foci or dipeptide repeats in the C9-expressing motor neurons was not evaluated. According to the results from C9-fly model, SRSF1 is participated in the multiple cell types such as neurons and glial cells. The role of the SRSF1 in the motor neurons should be evaluated. As aforementioned in our response to comment 1, we have now evaluated the role of SRSF1 depletion in neurons derived from patients (Fig. 8).

4. The authors' results suggest that cytoplasmic accumulation of C9 transcripts and dipeptides seems to be a key component of neurotoxicity. The referee feels it is worth discussing the results from other C9-ALS/FTD studies regarding this issue (e.x. Freibaum et al., Nature, 2015; Zhang et al., Nature, 2015, Jovicic et al., Nat Neurosci, 2015, Boeyanems et al., Sci Rep, 2016, Zhang et al., Nat Neurosci, 2016). On the other hand, in the other repeat-related diseases such as polyglutamine diseases or spinocerebellar ataxia, accumulation of the mutant products in nucleus seems to be toxic. Authors are encouraged to discuss the points. Our original manuscript was submitted in the format of a *Letter to Nature* and space constraint did not allow us to include all these references and discuss our findings in

relation to other microsatellite expansion disorders. We have now cited these references in the introduction and discussed this point.

5. Ethical statement is missing. Do the authors have an approval from institutional review board to perform the research using the patient-derived cells?

We forgot to include the ethical statement and apologise for this. Informed consent was obtained from all subjects before sample collection and our work is performed under Study number STH16573, Research Ethics Committee reference 12/YH/0330. This information has now been included in Methods in the section regarding the culture and differentiation of patient-derived cells.

Reviewer #3 (Remarks to the Author):

The manuscript by Hautbergue et al., proposes that interfering with the SRSF1-NXF1 nuclear export pathway prevents toxicity of the G4C2 and C4G2 repeats in flies and in cell culture by preventing nuclear export of the expansion RNAs. This hypothesis is largely based on previous data showing SRSF1 binds to G4C2 expansion transcripts in pull-down assays and in co-localization studies. Overall the authors show convincing evidence that RNAi knockdown of SRSF1 prevents neurodegeneration of fly eyes expressing 36 G4C2 repeats. Additionally, the authors show data that SRSF1 depletion suppresses C9orf72 astrocyte mediated toxicity - and that this decreased toxicity is associated with increased levels of nuclear RNA foci. In Figure 3 the authors extend these results using an overexpression strategy to express G4C2 and C4G2 antisense expansion RNAs. Again, cytoplasmic localization of the RNAs decreases SRSF1 depletion and similar effects are seen when SRSF1 mutations that block the interaction with NXF1 are expressed. While overall this is a strong manuscript, additional data are needed to support the central hypothesis, which is that SRSF1 binds to the C9 expansion RNAs and facilitates their cytoplasmic export. My specific comments follow:

1. Previous studies only provide evidence that the G4C2 expansion RNAs, not the C4G2 antisense RNAs bind to and co-localize with SRSF1. The antisense CCCCCG RNA has a very different motifs and structure compared to the GGGGCC repeat RNAs, which therefore may not bind to or sequester the same RNA binding protein. This raises questions about the model. The authors should test if CCCCCG binds to SRSF1 and also if it colocalizes with the antisense RNA foci. The best strategy to do this would be CLIP.

We fully agree with this point and have now performed two types of assay using *in vitro* UV-crosslinking reconstitution experiments and SRSF1 RNA immunoprecipitation (RIP) from formaldehyde-crosslinked neuronal Neuro-2a (N2A) cells transfected with sense or antisense repeat constructs bearing 15 or 38/39 hexanucleotide repeats.

Our data show that recombinant SRSF1 and ALYREF proteins purified from *E. coli* have the ability to directly interact with 5 repeats of G4C2-sense or C4G2-antisense synthetic radiolabeled RNA (Fig. 1a,b). Moreover, we now show sequestration-dependent length of SRSF1 on both sense and antisense repeat transcripts in live neuronal N2A cells (Fig. 6c,d).

We have tried to investigate the potential co-localization of SRSF1 with sense (in the *Brain* 2014 study) or antisense RNA foci in motor neurons from C9ORF72-ALS post-mortem tissues. We have indeed detected some nuclear co-localisation events in motor neurons from spinal cord and motor cortex. However, these might have been co-incident due to the diffuse nuclear staining of SRSF1. We have therefore preferred not to publish these data which remain inconclusive in our view. On the other hand, we consider that the potential co-localisation of SRSF1 with RNA foci is irrelevant in the context of nuclear export since RNA foci are unlikely to be exported through the nuclear pore due to their size. Single mRNA molecules require unfolding/unwinding by RNA helicases on both the nuclear and cytoplasmic sides for their passage through the nuclear pore. Moreover, while co-localisation may be a useful indicator of a functional

interaction, we have demonstrated both their physical interaction and a functional outcome by disrupting this interaction.

2. If binding of C4G2 expansion RNAs to SRSF1 does not occur then the central model does not fit well with the data as currently interpreted. If C4G2 RNAs do not bind SRSF1 it will be important to understand why these two different plasmids show similar results. We appreciate that this is a reasonable concern but we have now shown direct binding of SRSF1 onto synthetic antisense hexanucleotide repeat RNA *in vitro* (Fig 1b) and sequestration on repeat transcripts in live neuronal cells (Fig. 6d).

3. Since GP expression would occur from both sense and antisense directions, the authors should test if the reduction in toxicity and GP expression is caused by nuclear retention of antisense (G4C2 expansion RNAs expressed from C4G2 constructs). Strand specific RT-PCR or epitope tagged proteins could help determine this.

We have shown that nuclear export inhibition (or nuclear retention) of both sense or antisense constructs reduce the production of DPRs and associated cytotoxicity effects. Nuclear retention of both sense and antisense transcripts therefore decrease toxicity.

4. As mentioned, SRSF 1 is also a splicing factor. So it is important to show the C9orf72 intron 1 splicing pattern after applying SRSF1 RNAi in the iPS cells because the retention of the GGGGCC repeat containing intron will largely affect the nuclear-cytoplasmic localization of these repeats. Although SRSF 1 RNAi in cells transfected with GGGGCC constructs without splicing site reduced RAN translation, this does not exclude the possibility that both nuclear export and RNA splicing contribute to the cytoplasmic localization and RAN translation in patient derived cells. This should be addressed in the manuscript.

We fully agree with point. It was reported that the proportion of intron-1 spliced transcripts was not affected in C9ORF72-ALS post mortem brain tissue and neurons-derived from C9ORF72-ALS patient (Tran et al., 2015). We have obtained similar results in this study showing that the proportion of intron-1-spliced C9ORF72 transcripts is not affected in neurons-derived from C9ORF72-ALS patients compared to control patients (Fig. 8e,f). Moreover the depletion of SRSF1 does not affect the splicing of intron-1 in neurons derived from control or C9ORF72-ALS patients (Fig. 8e,f). On the other hand, while intron-1-retaining C9ORF72 transcripts were not detected in the neuronal cytoplasm of control patients in agreement with normal nuclear retention mechanisms, the presence of pathological repeat expansions in intron-1 of the C9ORF72 gene specifically licenses repeat transcripts for nuclear export (Fig. 8g,h). We further showed that the depletion SRSF1 specifically inhibits the nuclear export but not the expression levels (Fig. 8g,h) or intron-1-splicing of C9ORF72 repeat transcripts (Fig. 8e,f).

5. The authors tried to connect the NXF1 export receptor to the SRSF1 pathway by using the SRSF1-m4 point mutation, which does not interact with NXF1. Overexpression of this mutation leads to decreased cytoplasmic expansion RNAs and decreased RAN translation, but this could also due to the mutation affecting nuclear export or splicing activity of SRSF1. It is important to determine how the cytoplasmic expansion RNA and RAN protein levels are affected when knocking down NXF1 to fully clarify the pathway. We have indeed previously shown that the SRSF1-m4 mutation specifically inhibits the NXF1-dependent nuclear export of reporter mRNA in a structural and functional study (Tintaru et al., 2007). However, we fully agree that it would good to better validate our model suggesting that the RNA-repeat sequestration of SRSF1 triggers the NXF1-dependent nuclear export of C9ORF72-repeat transcripts in the neuronal context of our experiments.

We now show that the SRSF1-m4 fails to co-immunoprecipitate efficiently with endogenous NXF1 in neuronal N2A cells (Fig. 5e). Moreover, the SRSF1-m4 mutant directly interacts with 5 synthetic repeats of G4C2-sense or C4G2-antisense RNA in an *in vitro* reconstitution assay (Fig. 6a,b) and, similarly to SRSF1 wild-type, the SRSF1-m4 mutant has retained the ability to be sequestered on sense and antisense C9ORF72 repeat expanded mRNAs in live neuronal cells (Fig. 6c,d). Taken together, our data

clearly indicate that the sequestration on SRSF1-m4 onto repeat transcripts inhibits the recruitment of endogenous SRSF1 and the interaction with NXF1. We have also validated these findings in primary neurons (Fig. 7).

NXF1 is essential to the global nuclear export of mRNA and the knockdown of NXF1 leads to a severe block in the nuclear export of bulk mRNAs (Hautbergue et al., 2008; Williams et al., 2005) indicating that the nuclear export of most cellular mRNAs will be affected in cultured cells. This experiment will therefore not allow investigation of the specific impact of NXF1 on the direct nuclear export of *C9ORF72* repeat transcripts. However, consistent with the reviewer's idea, we found that the depletion of the NXF1 homologue in G4C2x36 *C9ORF72*-ALS flies restored locomotor deficits in larvae (Supplementary Fig. 6a) and adult flies (Supplementary Fig. 6b).

Figure 2 c is not clear as described and needs further explanation better images and a better description.

We have now provided enlarged images (Fig. 3c) and explained better the assay in the legend of the figure: "iAstrocytes-Motor Neurons co-cultures. iAstrocytes from controls and *C9ORF72*-ALS patients were transduced with Ad-RFP as viral control (A, B) and cultured with Hb9GFP+ motor neurons. As previously reported *C9ORF72*-ALS iAstrocytes exhibit toxicity against motor neurons (B). Transduction of Ad-RFP iAstrocytes with LV-*SRSF1*-RNAi co-expressing GFP led to consistent survival of Hb9GFP motor neurons on control astrocytes (C) and increase in survival on *C9ORF72*-ALS astrocytes (D)". We have also presented some high content imaging pictures showing how the *Columbus* analysis software recognizes motor neurons and the axons sprouting from the motor neuron perikarya over the astrocyte background (Supplementary fig. 2b).

Reviewer #4 (Remarks to the Author):

Hautbergue et al demonstrate that knockdown of SRSF1 suppresses toxicity of hexanucleotide repeat expression (G4C2 and C4G2) and suggest that this may be a therapeutic strategy for *C9ORF72*-related ALS. Using a *Drosophila* model, patient astrocyte/neuron cocultures, and transfected N2A cells, they conclude that SRSF1 is required for nuclear export of G4C2 and C2G4 RNA repeats. Finally, they use a dominant-negative SRSF1 construct to show that nuclear export of *C9ORF72* repeat transcripts and subsequent RAN translation depends on the interaction of SRSF1 with the NXF1 export receptor. If the data were convincing, this would be an extremely important finding. Unfortunately, though, the authors show limited data from multiple different model systems without many necessary controls, leaving the reader with an incomplete understanding of the effects of SRSF1 depletion in each system. The authors draw broad conclusions about mechanism from pieces of data obtained from very different model systems, and do not consider alternative explanations. SRSF1 regulates multiple aspects of RNA metabolism including transcription, stability, NMD, and translation. Importantly, an alternative explanation for suppression of their phenotypes in most experiments is a reduction in transcription or translation of the G4C2 repeats—while the authors quantitate RNA foci, they do not quantitate total message using RT-PCR, and RAN translation is only tested in an artificial system using transfected N2A cells.

In order to convince the reader that SRSF1 knockdown is truly a potential therapeutic strategy, the authors must test their hypothesis in a mammalian system in which the repeats are expressed in the context of the endogenous *c9orf72* gene. In this regard, experiments shown in figure 2d-e (using iAstrocytes and motor neuron co-cultures) come closest to testing their hypothesis, but the data provided are quite limited and unconvincing. First, how does SRSF1 knockdown in astrocytes prevent nonautonomous toxicity to motor neurons? This should be the ideal system to test whether SRSF1 knockdown inhibits DPR production (and presumably secretion into the media?), yet DPR production is not tested in this system.

This is a very nice suggestion, however, we have not been able to detect sufficient DPR

levels by western blots or DPR aggregates to allow quantification in astrocytes or neurons derived from patient despite numerous trials. We have not been able to detect sufficient levels of DPRs in the *Drosophila* model either.

We are not sure at this point how the inhibition of the nuclear export of *C9ORF72* repeat transcripts in astrocytes confers motor neuron neuroprotection. It might be that DPRs produced in astrocytes are packaged into the exosomal extra-cellular vesicles and exacerbate neurotoxicity when delivered to the motor neurons. Exosomes are also known to contain RNA molecules. It is also possible that the SRSF1 depletion affects the nuclear export and/or the expression of other RNAs involved in regulating neuronal homeostasis/death pathways. However, we consider that understanding the precise mechanisms by which *SRSF1* knockdown in astrocytes prevent non-autonomous toxicity to motor neurons is outside of the scope of this study.

Second, the authors state that they use this coculture assay to look at nonautonomous astrocyte-mediate toxicity because iNeurons from c9ALS patients do not have decreased survival. However, multiple groups have now shown that iPSC-derived neurons from c9-ALS patients have abnormal RNA profiles that mimic those seen in human tissue in addition to multiple cellular phenotypes (e.g. nuclear transport abnormalities) and increased sensitivity to stress. The authors should determine the effects of SRSF1 knockdown on neurons derived from c9orf72 patients and controls. An alternative approach if the authors cannot detect neuronal phenotypes in patient-derived iNeurons would be to look at the effect of SRSF1 knockdown on phenotypes recently described in BAC transgenic mice.

We now show that the depletion of SRSF1 in motor neurons derived from C9ORF72-ALS patients improve significantly their survival in co-cultures with astrocytes derived from C9ORF72-ALS patients (Fig. 8c) demonstrating a neuroprotective role of the SRSF1 depletion in the disease relevant cells.

We consider that gene profiling studies and SRSF1 depletion in BAC transgenic mice are out of the scope of this publication. Such experiments will be interesting to pursue in follow-up work, but would be likely to take at least a year to complete.

Another major question/concern is whether SRSF1 knockdown specifically inhibits G4C2 hexanucleotide RNA export or whether it has a global effect on mRNA export/transcription. The authors measure G4C2 mRNA export relative to U1 snRNA in Fig 3b in transfected N2A cells, but it would be important to perform similar experiments in iNeurons/astrocytes where they can determine the effect of SRSF1 knockdown on transcription/splicing/export of individual c9orf72 isoforms in addition to control genes. Does SRSF1 inhibit export of the entire c9orf72 mRNA with a retained G4C2-containing intron, or does it prevent export of spliced introns? At a minimum, the authors should measure total mRNA in nucleus vs cytoplasm as was performed in Friebaum et al, Nature 2015 (where the authors showed a general reduction in polyA+ export in C9 iPSCs).

We agree with this comment and have now performed the total, nuclear and cytoplasmic analysis of intron-1-spliced and intron-1-retaining *C9ORF72* transcripts in neurons derived from control and C9ORF72-ALS patients treated or not with the SRSF1-RNAi lentivirus. We now show that the proportion of intron-1-spliced *C9ORF72* transcripts is not affected in neurons-derived from C9ORF72-ALS patients compared to control patients (Fig. 8e,f) in full agreement with a previous study from the Gao group (Tran et al., 2015). Moreover the depletion of SRSF1 does not affect the splicing of intron-1 in neurons derived from control or C9ORF72-ALS patients (Fig. 8e,f). On the other hand, while intron-1-retaining *C9ORF72* transcripts were not detected in the neuronal cytoplasm of control individuals, the presence of pathological repeat expansions in intron-1 of the *C9ORF72* gene specifically licenses repeat transcripts for nuclear export (Fig. 8g,h). We further showed that the depletion of SRSF1 specifically inhibits the nuclear export but not the expression levels (Fig. 8g,h) or intron-1-splicing of *C9ORF72* repeat transcripts (Fig. 8e,f). Total levels of intron-1-spliced and intron-1-retaining transcripts are not affected by SRSF1 depletion, ruling out major effects of *SRSF1* depletion on the transcription or the stability

of *C9ORF72* repeat transcripts (Fig. 8e,h). Similarly, we also show that the total levels of *C9ORF72* repeat transcripts are not affected in *G4C2x36+SRSF1-RNAi Drosophila* (Supplementary Fig. 7b) while the cytoplasmic levels were markedly reduced (Fig. 6h).

Moreover, the analysis of the effects of SRSF1 knockdown in *Drosophila* is very superficial. There are many simple yet critical experiments that were omitted.

1) A critical control experiment is to determine if SRSF1 knockdown suppresses toxicity caused by overexpression of DPRs (“protein only “ GR or PR). The authors claim to show in extended data fig. 7 that SRSF1-RNAi doesn’t significantly suppress the climbing phenotype caused by Arginine-containing DPRs even though there does appear to be some rescue (but doesn’t reach statistical significance—surprisingly, they only test 8 flies for GR36!). I suspect with increasing N (to level of control), they will see significant rescue. An easier experiment, though, is just to look for suppression of the rough eye phenotype caused by GR36 and PR36. If SRSF1-RNAi suppresses the GR36 and PR36 rough eye phenotype, this would suggest a general inhibition of transcription/export rather than being specific to G4C2 repeats.

The reviewer will no doubt appreciate that the expression of arginine-containing repeats is highly toxic, as elaborated by Mizieliński et al (2014), especially so for the GR repeats. We repeated this experiment to generate additional flies to test. However, despite concerted effort we were able to recover very few GR36 expressing adult flies, which again performed very poorly in our climbing assay, but we were unable to recover additional flies expressing GR36 with SRSF1-RNAi (this itself indicates no improvement in viability). Nevertheless, our climbing assay has been highly optimized to be able to detect even subtle differences between groups with the animals we are able to test. While the low number for GR36 should be interpreted cautiously, there is no statistical difference between groups for GR36 or even PR36 where numbers were not limiting. However, to extend these observations as the reviewer suggested, we have analyzed the effect of *SRSF1-RNAi* on the rough eye phenotype induced by GR36 and PR36. In agreement with Mizielińska et al. GR36 gives a very pronounced rough eye while PR36 generates a milder rough eye phenotype. In contrast to the suppression of G4C2x36 induced rough eye, *SRSF1-RNAi* does not modify the rough eye phenotype of either GR36 or PR36. Importantly, in all of these experiments, we have been careful to balance the number of UAS transgenes so as to control for non-specific effects of titrating the induction of the *C9ORF72* model expression. In conclusion, we maintain that *SRSF1-RNAi* does not affect toxicity induced by transgenic expression of G4C2-independent DPRs.

We thank the reviewer for suggesting that we extend this part of the study and we appreciate its importance, so we have now included these data in Figure 2e,f.

2) The fly model used has serious caveats when it comes to analysis of G4C2 RNA export since it contains a polyA tail after the repeats. To show that SRSF1 knockdown specifically inhibits G4C2 export (rather than globally reducing transcription or export), the authors should assess G4C2 levels (total, nuclear, and cytoplasmic) in addition to that of control mRNAs. They should also test the effects in the “intron model” developed by the Gao lab in which they show correlation of toxicity with nuclear export and RAN translation.

This is a good point and we now show that the total levels of *C9ORF72* repeat transcripts are not affected in *G4C2x36+SRSF1-RNAi Drosophila* (Supplementary Fig. 7b) while the cytoplasmic levels were markedly reduced (Supplementary Fig. 7b), consistent with data obtained in N2A cells (Supplementary Fig. 7a). We also show the cytoplasmic/total ratio in Fig. 6h (for *Drosophila* assay) and Fig. 6f (for N2A cells assay). These data provide further evidence that *SRSF1-RNAi* specifically reduces the cytoplasmic amount of G4C2 transcripts.

As suggested by the reviewer we obtained the “intron model” lines from Prof. Gao. In their original study, Tran et al. reported very little phenotypic effect from 160 G4C2 repeats (160R) expressed in the ‘intronic’ context of *C9ORF72*, in contrast to DPR or ‘polyA’ G4C2 repeat transgenes. However, they did report a modest effect on lifespan when 160R intronic repeats were expressed at elevated levels at 29°C using the motor

neuron driver OK371-GAL4. We set up the same experiment with the intention of testing whether the *SRSF1*-RNAi can also ameliorate this phenotype. However, despite using the exact driver and responder lines, using exactly the same experimental set up as described, we were unable to replicate the lifespan defect by OK371>160R expression at 29°C. These data are shown below for the reviewer. We are in the process of repeating this on the off-chance that this result was erroneous, but at 21 days we still do not observe any effect of 160R above background. Tran et al. reported a marked decline by ~15 days. Therefore, while we appreciate this as a good suggestion, we have not yet found conditions in which the “intronic” model produces a phenotype with which we can test our hypothesis. On reflection, Tran et al. did describe that their model produces poly-GP at 100-fold less than 36-repeat ‘polyA’ transgenes, which accounts for the lack of toxicity.

Finally, the authors propose that the mechanism by which the G4C2 repeats export the nucleus is via a direct interaction with SRSF1. This model would lead to several predictions. First, increasing SRSF1 should enhance export and therefore toxicity. Overexpression of SRSF1 did not increase the yield of DPRs in transfected Neuro2A (Fig. 5f). In fact, bulk mRNA nuclear export is blocked in cells overexpressing nuclear export adaptors (Chang et al., 2013; Hautbergue et al., 2009) as free nuclear export adaptors are able to interact with NXF1 in the absence of co-transcriptionally processing mRNA. However, the high affinity remodeling of NXF1 required for the nuclear export of mRNA requires both interactions with the nuclear export adaptor and the RNA. Consistent with this, overexpression of SRSF1 was also reported to be toxic in *Drosophila* possibly due to an inhibition of the bulk nuclear export of mRNA (Allemand et al., 2001).

Second, mutating SRSF1 to specifically prevent interaction with G4C2 repeats should also suppress export and toxicity. These experiments should be performed to test their proposed mechanism.

On the contrary, we think that expression of a SRSF1 mutant that does not bind repeat transcripts would not lead to inhibition of nuclear export and toxicity as the endogenous SRSF1 will still be sequestered on the repeat transcripts and trigger nuclear export via the NXF1-dependent pathway. However, sequestration of the SRSF1-m4 mutant which fails to interact with NXF1 and prevents the recruitment of endogenous SRSF1 onto repeat hexanucleotide transcripts (Fig. 6a,d) indeed acts as a dominant negative mutant reducing the nuclear export of repeat transcripts (Fig. 6f) and also DPR production (Fig. 5f).

Other considerations:

The submission appears to be a Nature letter format with only 3 figures rather than a Nature Communications article format.

As the reviewer has noted this manuscript was transferred directly from Nature to Nature

Communications without an opportunity to reformat. The manuscript has now been appropriately reformatted for Nature Communications.

Extended data Figure 1 and 5 are almost identical – (human vs mouse). One extended figure to show generation of this construct is more than sufficient.

We have merged these two figures (Supplementary Fig. 1a,b).

Minor points:

Figure 2c legend – authors should state what green fluorescence is (Hb9-GFP+ MNs).

This has been added to the figure legend.

References

Allemand, E., Gattoni, R., Bourbon, H.M., Stevenin, J., Cáceres, J.F., Soret, J., and Tazi, J. (2001). Distinctive features of *Drosophila* alternative splicing factor RS domain: implication for specific phosphorylation, shuttling, and splicing activation. *Molecular and Cellular Biology* *21*, 1345–1359.

Chang, C.-T., Hautbergue, G.M., Walsh, M.J., Viphakone, N., van Dijk, T.B., Philipsen, S., and Wilson, S.A. (2013). Chtop is a component of the dynamic TREX mRNA export complex. *Embo J.* *32*, 473–486.

Cooper-Knock, J., Higginbottom, A., Stopford, M.J., Highley, J.R., Ince, P.G., Wharton, S.B., Pickering-Brown, S., Kirby, J., Hautbergue, G.M., and Shaw, P.J. (2015). Antisense RNA foci in the motor neurons of. *Acta Neuropathol* *130*, 63–75.

Cooper-Knock, J., Walsh, M.J., Higginbottom, A., Robin Highley, J., Dickman, M.J., Edbauer, D., Ince, P.G., Wharton, S.B., Wilson, S.A., Kirby, J., et al. (2014). Sequestration of multiple RNA recognition motif-containing proteins by C9orf72 repeat expansions. *Brain* *137*, 2040–2051.

Donnelly, C.J., Zhang, P.-W., Pham, J.T., Heusler, A.R., Mistry, N.A., Vidensky, S., Daley, E.L., Poth, E.M., Hoover, B., Fines, D.M., et al. (2013). RNA Toxicity from the ALS/FTD C9ORF72 Expansion Is Mitigated by Antisense Intervention. *Neuron* *80*, 415–428.

Hautbergue, G.M., Hung, M.-L., Golovanov, A.P., Lian, L.-Y., and Wilson, S.A. (2008). Mutually exclusive interactions drive handover of mRNA from export adaptors to TAP. *Proc. Natl. Acad. Sci. U.S.A.* *105*, 5154–5159.

Hautbergue, G.M., Hung, M.-L., Walsh, M.J., Snijders, A.P.L., Chang, C.-T., Jones, R., Ponting, C.P., Dickman, M.J., and Wilson, S.A. (2009). UIF, a New mRNA export adaptor that works together with REF/ALY, requires FACT for recruitment to mRNA. *Curr. Biol.* *19*, 1918–1924.

Katahira, J., Inoue, H., Hurt, E., and Yoneda, Y. (2009). Adaptor Aly and co-adaptor Thoc5 function in the Tap-p15-mediated nuclear export of HSP70 mRNA. *Embo J.* *28*, 556–567.

Kwon, I., Xiang, S., Kato, M., Wu, L., Theodoropoulos, P., Wang, T., Kim, J., Yun, J., Xie, Y., and McKnight, S.L. (2014). Poly-dipeptides encoded by the C9ORF72 repeats bind nucleoli, impede RNA biogenesis, and kill cells. *Science*.

Lee, Y.-B., Chen, H.-J., Peres, J.N., Gomez-Deza, J., Attig, J., Štálekár, M., Troakes, C., Nishimura, A.L., Scotter, E.L., Vance, C., et al. (2013). Hexanucleotide Repeats in ALS/FTD Form Length-Dependent RNA Foci, Sequester RNA Binding Proteins, and Are Neurotoxic. *Cell Rep* *5*, 1178–1186.

Masuda, S., Das, R., Cheng, H., Hurt, E., Dorman, N., and Reed, R. (2005). Recruitment of the human TREX complex to mRNA during splicing. *Genes & Development* 19, 1512–1517.

Meyer, K., Ferraiuolo, L., Miranda, C.J., Likhite, S., McElroy, S., Rensch, S., Ditsworth, D., Lagier-Tourenne, C., Smith, R.A., Ravits, J., et al. (2014). Direct conversion of patient fibroblasts demonstrates non-cell autonomous toxicity of astrocytes to motor neurons in familial and sporadic ALS. *Proc. Natl. Acad. Sci. U.S.A.* 111, 829–832.

Mizielinska, S., Grönke, S., Niccoli, T., Ridler, C.E., Clayton, E.L., Devoy, A., Moens, T., Norona, F.E., Woollacott, I.O.C., Pietrzyk, J., et al. (2014). C9orf72 repeat expansions cause neurodegeneration in *Drosophila* through arginine-rich proteins. *Science* 345, 1192–1194.

Müller-McNicoll, M., Botti, V., de Jesus Domingues, A.M., Brandl, H., Schwich, O.D., Steiner, M.C., Curk, T., Poser, I., Zarnack, K., and Neugebauer, K.M. (2016). SR proteins are NXF1 adaptors that link alternative RNA processing to mRNA export. *Genes & Development* 30, 553–566.

Niblock, M., Smith, B.N., Lee, Y.-B., Sardone, V., Topp, S., Troakes, C., Al-Sarraj, S., Leblond, C.S., Dion, P.A., Rouleau, G.A., et al. (2016). Retention of hexanucleotide repeat-containing intron in C9orf72 mRNA: implications for the pathogenesis of ALS/FTD. *Acta Neuropathol Commun* 4, 18.

Sareen, D., O'Rourke, J.G., Meera, P., Muhammad, A.K.M.G., Grant, S., Simpkinson, M., Bell, S., Carmona, S., Ornelas, L., Sahabian, A., et al. (2013). Targeting RNA Foci in iPSC-Derived Motor Neurons from ALS Patients with a C9ORF72 Repeat Expansion. *Sci Transl Med* 5, 208ra149.

Strässer, K., Masuda, S., Mason, P., Pfannstiel, J., Oppizzi, M., Rodríguez-Navarro, S., Rondón, A.G., Aguilera, A., Struhl, K., Reed, R., et al. (2002). TREX is a conserved complex coupling transcription with messenger RNA export. *Nature* 417, 304–308.

Tintaru, A.M., Hautbergue, G.M., Hounslow, A.M., Hung, M.-L., Lian, L.-Y., Craven, C.J., and Wilson, S.A. (2007). Structural and functional analysis of RNA and TAP binding to SF2/ASF. *EMBO Rep.* 8, 756–762.

Tran, H., Almeida, S., Moore, J., Gendron, T.F., Chalasani, U., Lu, Y., Du, X., Nickerson, J.A., Petrucelli, L., Weng, Z., et al. (2015). Differential Toxicity of Nuclear RNA Foci versus Dipeptide Repeat Proteins in a *Drosophila* Model of C9ORF72 FTD/ALS. *Neuron* 87, 1207–1214.

Viphakone, N., Hautbergue, G.M., Walsh, M., Chang, C.-T., Holland, A., Folco, E.G., Reed, R., and Wilson, S.A. (2012). TREX exposes the RNA-binding domain of Nxf1 to enable mRNA export. *Nat Commun* 3, 1006.

Walsh, M.J., Hautbergue, G.M., and Wilson, S.A. (2010). Structure and function of mRNA export adaptors. *Biochem. Soc. Trans.* 38, 232–236.

Williams, B.J.L., Boyne, J.R., Goodwin, D.J., Roaden, L., Hautbergue, G.M., Wilson, S.A., and Whitehouse, A. (2005). The prototype gamma-2 herpesvirus nucleocytoplasmic shuttling protein, ORF 57, transports viral RNA through the cellular mRNA export pathway. *Biochem. J.* 387, 295–308.

Reviewers' comments:

Reviewer #1 (Remarks to the Author):

The authors have improved their original manuscript by showing that SRSF1 depletion does not alter total C9ORF72 transcript levels or splicing and have included studies in primary neurons and iNeurons. The amount of work that has gone into these additional studies is acknowledged, and the finding of SRSF1 binding to G4C2 expansions and the mechanism leading to export of C9ORF72 repeat transcripts is very original, but there are two major concerns with the study.

1. From the text

'P4, 2nd para, line 10: In this study, we demonstrate that sequestration of SRSF1 onto C9ORF72 repeat transcripts triggers their NXF1-related nuclear export independent of splicing and the subsequent production of neurotoxic levels of DPRs by RAN-translation in the cytoplasm.'

The premise from the above and from the argument in the manuscript is that SRSF1 binds to G4C2 and C4G2 repeat expansions, which recruits NXF1 leading to export of C9ORF72 repeat transcripts to the cytoplasm where they are translated to generate toxic dipeptide repeat proteins (DPRs). Downregulation of SRSF1 retains the C9ORF72 repeat transcripts in the nucleus thereby lowering DPRs and preventing toxicity.

The major concern with this is that in the key models used to test this hypothesis changes in DPR expression are not shown, indeed DPRs are not even detected. For example, depletion of SRSF1 alleviates the disease phenotype in G4C2x36 Drosophila model, but since DPRs are not detected (as pointed out by the authors in the rebuttal, P10) the underlying mechanism is unclear.

Similarly in the Hb9-GFP+ mouse motor neurons/iAstrocyte co-culture experiments (Figure 3), DPRs are not detected in the C9ORF72-ALS astrocytes, and indeed the authors point out that they do not know the mechanism by which SRSF1 depletion in C9ORF72-ALS astrocytes reduces toxicity to the Hb9-GFP+ mouse motor neurons, citing possible changes in exosome cargoes or that SRSF1 depletion affects the nuclear export and/or the expression of other RNAs involved in regulating neuronal homeostasis/death pathways. This is broad speculation and does not support their conclusion that SRSF1 depletion causes nuclear retention of C9ORF72 repeat transcripts leading to a reduction in toxic DPRs.

This also applies to the experiments using co-cultures of C9ORF72-ALS iMNs and C9ORF72-ALS astrocytes, where i) DPRs are not assessed/detected ii) changes in RNA foci (less in cytoplasm more in the nucleus) are not shown; iii) and it is not clear if the SRSF1-RNAi treatments were applied separately to the neuronal cultures then co-cultured with the astrocytes, so the targeting of the SRSF1-RNAi is unclear.

2. The authors were asked to show co-localization of SRSF1 to G4C2/C4G2 RNA foci in disease tissues to strengthen the relevance of their finding to the disease mechanism. The authors state in their rebuttal that 'we consider that the potential co-localisation of SRSF1 with RNA foci is irrelevant in the context of nuclear export since RNA foci are unlikely to be exported through the nuclear pore due to their size' however the authors have published an entire paper looking at colocalization of candidate proteins (ALYREF, SRSF2, hnRNP h1/F and SRSF1) and C9ORF72 RNA foci using combined RNA FISH and immunohistochemistry to support their in vitro findings (Cooper-Knock Brain 2014), so clearly these studies have relevance. If as the authors state SRSF1 would not necessarily be co-localized with RNA foci, then this suggests SRSF1 would bind only to transcripts destined for export, this would need to be proven.

Reviewer #2 (Remarks to the Author):

The revised manuscript has been improved by addressing many of the previous points raised by the reviewers. The comments on the revised version are provided as follows.

1. Fig 8; The specific effect of SRSF1 to the repeat-containing C9orf72 on the nuclear retention was demonstrated in the patient-derived neurons. This supports their hypothesis of the cytoplasmic rather than nuclear DPR being toxic. However, DPR protein was not detected. The proof by the protein level in the cytoplasmic fraction is important.
2. Besides Tran and colleague (2015), are there any other published evidence to support the authors' hypothesis? Still, in most of the repeat-related diseases such as polyglutamine diseases or spinocerebellar ataxia, accumulation of the mutant products in nucleus is central to the toxicity.
3. Figure 3, 8: Authors showed neuroprotective effect of SRSF inhibition in C9orf72 patient-derived neurons. The effect seems more robust in manipulation of C9-astrocytes (Fig 3d) than C9-neurons. Do authors think which cells are central to the C9-mediated toxicities? In addition, the mechanism through which non-cell autonomous toxicity by C9-astrocytes is not clear.

Reviewer #4 (Remarks to the Author):

This revised manuscript is dramatically improved. The authors have now convinced me of the importance of their findings, and I believe this will be a key advancement in the field. They have satisfactorily addressed all my concerns, many of which were shared by other reviewers. I believe they have made a valiant attempt at satisfying all the reviewers' concerns.

Point-by-point response to reviewers for article NCOMMS-16-13700A

Our responses to reviewers' comments are labeled in blue.

Reviewers' comments:

Reviewer #1 (Remarks to the Author):

The authors have improved their original manuscript by showing that SRSF1 depletion does not alter total C9ORF72 transcript levels or splicing and have included studies in primary neurons and iNeurons. The amount of work that has gone into these additional studies is acknowledged, and the finding of SRSF1 binding to G4C2 expansions and the mechanism leading to export of C9ORF72 repeat transcripts is very original, but there are two major concerns with the study.

1. From the text

'P4, 2nd para, line 10: In this study, we demonstrate that sequestration of SRSF1 onto C9ORF72 repeat transcripts triggers their NXF1-related nuclear export independent of splicing and the subsequent production of neurotoxic levels of DPRs by RAN-translation in the cytoplasm.'

The premise from the above and from the argument in the manuscript is that SRSF1 binds to G4C2 and C4G2 repeat expansions, which recruits NXF1 leading to export of C9ORF72 repeat transcripts to the cytoplasm where they are translated to generate toxic dipeptide repeat proteins (DPRs). Downregulation of SRSF1 retains the C9ORF72 repeat transcripts in the nucleus thereby lowering DPRs and preventing toxicity.

The major concern with this is that in the key models used to test this hypothesis changes in DPR expression are not shown, indeed DPRs are not even detected. For example, depletion of SRSF1 alleviates the disease phenotype in G4C2x36 *Drosophila* model, but since DPRs are not detected (as pointed out by the authors in the rebuttal, P10) the underlying mechanism is unclear.

We appreciate that this is an important point and redoubled our efforts to detect DPRs *in vivo*. We are now not only able to detect DPRs from *Drosophila* tissue but also to show that the depletion of SRSF1 leads to prominent reduction of sense and antisense poly-GP DPRs using same detection assays (dot blots) as in the original publication that described the C9ORF72-ALS *Drosophila* model and the fact that DPRs are primarily implicated in C9ORF72 repeat transcripts mediated neurotoxicity (Mizielinska S et al. *Science* 2014; **345**:1192-4). Consistently, our new data show that partial depletion of SRSF1 leads to prominent reduction of DPRs and confers in turn neuroprotection. The new data have been included as a new Supplementary Figure 2a.

Similarly in the Hb9-GFP+ mouse motor neurons/iAstrocyte co-culture experiments (Figure 3), DPRs are not detected in the C9ORF72-ALS astrocytes, and indeed the authors point out that they do not know the mechanism by which SRSF1 depletion in C9ORF72-ALS astrocytes reduces toxicity to the Hb9-GFP+ mouse motor neurons, citing possible changes in exosome cargoes or that SRSF1 depletion affects the nuclear export and/or the expression of other RNAs involved in regulating neuronal homeostasis/death pathways. This is broad speculation and does not support their conclusion that SRSF1 depletion causes nuclear retention of C9ORF72 repeat transcripts leading to a reduction in toxic DPRs.

As agreed by the editor, the precise molecular mechanisms by which SRSF1 depletion in C9ORF72-ALS astrocytes reduces toxicity to the Hb9-GFP+ mouse motor neurons falls out of the scope of this study. We have however modulated our text to avoid undue

speculation.

This also applies to the experiments using co-cultures of C9ORF72-ALS iMNs and C9ORF72-ALS astrocytes, where i) DPRs are not assessed/detected ii) changes in RNA foci (less in cytoplasm more in the nucleus) are not shown; iii) and it is not clear if the SRSF1-RNAi treatments were applied separately to the neuronal cultures then co-cultured with the astrocytes, so the targeting of the SRSF1-RNAi is unclear.

We have now clarified in the text that the *SRSF1*-RNAi was applied to motor neurons prior to the co-culture assays. The text has been modified in the result section on page 14 and in the discussion on page 15.

Furthermore, we are now able to show reduced production of sense and antisense poly-GP DPRs in SRSF1-RNAi transduced motor neurons derived from C9ORF72-ALS patients. The new data have been included as a new Supplementary Figure 2b. This validates completely our model in patient-derived motor neurons and is in full agreement with our previous experiments showing that partially depleting SRSF1 or inhibiting its sequestration on repeat RNA and interaction with the nuclear export machinery lead to altered production of DPRs in both N2A neuronal cell models and primary neurons.

2. The authors were asked to show co-localization of SRSF1 to G4C2/C4G2 RNA foci in disease tissues to strengthen the relevance of their finding to the disease mechanism. The authors state in their rebuttal that 'we consider that the potential co-localisation of SRSF1 with RNA foci is irrelevant in the context of nuclear export since RNA foci are unlikely to be exported through the nuclear pore due to their size' however the authors have published an entire paper looking at colocalization of candidate proteins (ALYREF, SRSF2, hnRNP h1/F and SRSF1) and C9ORF72 RNA foci using combined RNA FISH and immunohistochemistry to support their in vitro findings (Cooper-Knock Brain 2014), so clearly these studies have relevance. If as the authors state SRSF1 would not necessarily be co-localized with RNA foci, then this suggests SRSF1 would bind only to transcripts destined for export, this would need to be proven.

Using confocal immunofluorescence microscopy, we show co-localization of SRSF1 with RNA foci in post-mortem C9ORF72-related ALS human motor neurons and in disease-relevant tissue (spinal cord). The new data have been included as a new panel in figure 1 (panel c).

Reviewer #2 (Remarks to the Author):

The revised manuscript has been improved by addressing many of the previous points raised by the reviewers. The comments on the revised version are provided as follows.

1. Fig 8; The specific effect of SRSF1 to the repeat-containing C9orf72 on the nuclear retention was demonstrated in the patient-derived neurons. This supports their hypothesis of the cytoplasmic rather than nuclear DPR being toxic. However, DPR protein was not detected. The proof by the protein level in the cytoplasmic fraction is important.

We now show reduced production of sense and antisense poly-GP DPRs in SRSF1-RNAi transduced motor neurons derived from C9ORF72-ALS patients. The new data have been included as a new Supplementary Figure 2b. This validates completely our

model in patient-derived motor neurons and is in full agreement with our previous experiments showing that partially depleting SRSF1 or inhibiting its sequestration on repeat RNA and interaction with the nuclear export machinery lead to altered production of DPRs in both N2A neuronal cell models and primary neurons.

The reviewer states "This supports their hypothesis of the cytoplasmic rather than nuclear DPR being toxic". We disagree with this suggestion. DPRs are imported in the nucleus and can therefore be found in both the nucleus and the cytoplasm. Our study does not investigate the cellular localisation of RAN-translated DPRs or the nucleocytoplasmic transport of DPRs. It focuses on the nuclear export of *C9ORF72* repeat RNA transcripts, which are subsequently translated into DPRs and shuttle between the nucleus and the cytoplasm. Our model predicts that the SRSF1-dependent nuclear export inhibition of *C9ORF72* repeat RNA transcripts and the reduced cytoplasmic levels of *C9ORF72* repeat transcripts, that have been quantified in three models (*Drosophila*, patient derived neurons and neuronal cell models), lead to the subsequent inhibition of global expression of DPRs. This is exactly what we are now showing in the same exact three models, completely confirming the predictions of our model both *in vivo* in *Drosophila* and *in vitro* in patient-derived neurons and neuronal cell models.

2. Besides Tran and colleague (2015), are there any other published evidence to support the authors' hypothesis? Still, in most of the repeat-related diseases such as polyglutamine diseases or spinocerebellar ataxia, accumulation of the mutant products in nucleus is central to the toxicity.

On the contrary, there is a growing body of evidence to support the neurotoxic role of DPRs both in *in vivo* and *in vitro* models. References below are examples of articles that support this concept and have been published in leading journals, and is by no means an exhaustive list:

- Zu, T. et al. RAN proteins and RNA foci from antisense transcripts in *C9ORF72* ALS and frontotemporal dementia. *Proc. Natl. Acad. Sci. U.S.A.* 110, E4968–77 (2013).

- Mori, K. et al. The *C9orf72* GGGGCC repeat is translated into aggregating dipeptide-repeat proteins in FTL/ALS. *Science* 339, 1335–1338 (2013).

- Mori, K. et al. Bidirectional transcripts of the expanded *C9orf72* hexanucleotide repeat are translated into aggregating dipeptide repeat proteins. *Acta Neuropathol* 126, 881–893 (2013).

- Ash, P. E. A. et al. Unconventional translation of *C9ORF72* GGGGCC expansion generates insoluble polypeptides specific to c9FTD/ALS. *Neuron* 77, 639–646 (2013).

- Kwon I et al. Poly-dipeptides encoded by the *C9orf72* repeats bind nucleoli, impede RNA biogenesis, and kill cells. *Science* 2014; 345:1139-45.

- Mizielińska, S. et al. *C9orf72* repeat expansions cause neurodegeneration in *Drosophila* through arginine-rich proteins. *Science* 345, 1192–1194 (2014).

- Lee KH et al. *C9orf72* Dipeptide Repeats Impair the Assembly, Dynamics, and Function of Membrane-Less Organelles. *Cell* 2016; 167:774-88.

- Boeynaems S et al. *Mol Cell*. Phase Separation of *C9orf72* Dipeptide Repeats Perturbs Stress Granule Dynamics. 2017; 65, 1044-55.

We agree with the reviewer that neurotoxicity in polyglutamine diseases, fragile X-associated tremor ataxia syndrome (FXTAS), spinocerebellar ataxia or myotonic dystrophy was originally attributed to the nuclear accumulation of RNA foci and RNA-repeat sequestration of proteins, particularly the splicing factors MBLN1, Sam68, PUR-alpha or CUGBP1 (Greco C et al. Brain 2006, 129:243-55; Sellier C et al. EMBO J. 2010; 29:248-61; Jin P et al. Neuron 2007; 55:556-64; Sofola OA et al. Neuron 2007; 55:565-71; Li L-B et al. Nature 2008; 453:1107-11; Tsoi H et al. Hum. Mol. Genet. 2011; 20:3787-97). However, RAN translation was not yet discovered (Zu et al. Proc. Natl. Acad. Sci. U.S.A. 2011; 108:260-5). Interestingly, in FXTAS for example, the discovery of RAN translation in the same model as in (Jin P et al. Neuron 2007; 55:556-64; Sofola OA et al. Neuron 2007) challenged the view that neurotoxicity is primarily caused by RNA foci and RNA-repeat sequestration of splicing factors (Todd PK et al. 2013; 78:440-55). Similarly, in C9ORF72-ALS, increasing 10 fold the number of intranuclear RNA foci does not significantly alter cell survival or global RNA processing while expression of DPRs cause neurodegeneration (Tran H et al. Neuron 2015; 87:1207-14). We have added a paragraph about this interesting debate in the discussion during the first round of revision.

3. Figure 3, 8: Authors showed neuroprotective effect of SRSF inhibition in C9orf72 patient-derived neurons. The effect seems more robust in manipulation of C9-astrocytes (Fig 3d) than C9-neurons. Do authors think which cells are central to the C9-mediated toxicities? In addition, the mechanism through which non-cell autonomous toxicity by C9-astrocytes is not clear.

As agreed by the editor, the precise molecular mechanisms by which SRSF1 depletion in C9ORF72-ALS astrocytes reduces non-cell autonomous toxicity to motor neurons falls out of the scope of this study.

Reviewer #4 (Remarks to the Author):

This revised manuscript is dramatically improved. The authors have now convinced me of the importance of their findings, and I believe this will be a key advancement in the field. They have satisfactorily addressed all my concerns, many of which were shared by other reviewers. I believe they have made a valiant attempt at satisfying all the reviewers' concerns.

We are pleased to have fully addressed the reviewer's comments in our first revision and acknowledge the reviewer for this very supportive statement.

REVIEWERS' COMMENTS:

Reviewer #1 (Remarks to the Author):

The authors have addressed all comments, just two slight modifications.

1) Supplementary Figure 2a

Showing that SRSF1 downregulation lowers DPRs is a key piece of data and should be included in the main text, not the supplemental. Patient 201 looks convincing, but patient 78 does not.

2) Abstract Line 3-4

'Expression of repeat transcripts and dipeptide-repeat proteins by unconventional translation in the cytoplasm leads to progressive death of motor neurons.'

'leads to progressive death of motor neurons' is too strong a statement since the toxicity of dipeptide-repeat proteins in motor neurons remains controversial.

Reviewer #2 (Remarks to the Author):

The revised manuscript has been significantly improved by addressing many of the previous points raised by the reviewers.

Significant reduction of a level of DPR upon siRNA for SRSF1 was demonstrated in drosophila larvae, with a modest reduction of DPR upon siRNA for SRSF1 being demonstrated in human induced motor neurons.

With additional data provided in this version, this reviewer is satisfied with the current version of the manuscript.

Point-by-point response to reviewers for article NCOMMS-16-13700C

We acknowledge the reviewers for their positive comments. We have incorporated the requested modifications in the revised manuscript. Our response is labeled in blue.

REVIEWERS' COMMENTS:

Reviewer #1 (Remarks to the Author):

The authors have addressed all comments, just two slight modifications.

1) Supplementary Figure 2a

Showing that SRSF1 downregulation lowers DPRs is a key piece of data and should be included in the main text, not the supplemental. Patient 201 looks convincing, but patient 78 does not.

We have now included the Supplementary Figure 2 into the main figures. Panel a was inserted in the *Drosophila* figure (Fig. 2e) and panel b was inserted in the patient-derived induced motor neurons figure (Fig. 8d). The editor requested moving the entire Supplementary Figure 2 into the main figures. A few sentences were added in the result section and in the discussion to highlight variability in the SRSF1-RNAi-dependent reduction of DPRs production between patient-derived motor neurons.

2) Abstract Line 3-4

'Expression of repeat transcripts and dipeptide-repeat proteins by unconventional translation in the cytoplasm leads to progressive death of motor neurons.'

'leads to progressive death of motor neurons' is too strong a statement since the toxicity of dipeptide-repeat proteins in motor neurons remains controversial.

The sentence has been reverted to the text used in the first round of revision and which all reviewers had accepted. The sentence now reads "Expression of repeat transcripts and dipeptide-repeat proteins trigger multiple mechanisms of neurotoxicity".

Reviewer #2 (Remarks to the Author):

The revised manuscript has been significantly improved by addressing many of the previous points raised by the reviewers.

Significant reduction of a level of DPR upon siRNA for SRSF1 was demonstrated in drosophila larvae, with a modest reduction of DPR upon siRNA for SRSF1 being demonstrated in human induced motor neurons.

With additional data provided in this version, this reviewer is satisfied with the current version of the manuscript.

We acknowledge this positive comment.